# Unveiling Over-Memorization in Finetuning LLMs for Reasoning Tasks

## Abstract

The pretrained large language models (LLMs) are finetuned with labeled data for better instruction following ability and alignment with human values. In this paper, we study the learning dynamics of LLM finetuning on reasoning tasks and reveal the uncovered over-memorization phenomenon during a specific stage of LLM finetuning. At this stage, the LLMs have excessively memorized training data and exhibit high test perplexity while maintaining good test accuracy. We explore the conditions that contribute to over-memorization and discover that this issue is prevalent across various tasks, models, and fine-tuning methods, with prolonged training and large learning rates exacerbating the problem. Although models with over-memorization demonstrate comparable test accuracy to normal models, they suffer from reduced robustness, poor out-of-distribution generalization, and decreased generation diversity. In light of our findings on over-memorization, we offer recommendations for checkpoint selection and propose techniques such as checkpoint merging and memorization-aware reweighting to mitigate this effect.

## 1 Introduction

Large language models (LLMs) demonstrate remarkable capabilities attributed to the expansion of both training data and model parameters (Fedus et al., 2022; Achiam et al., 2023; AI@Meta, 2024; Team, 2024b; Brown et al., 2020). To adapt these models to domain-specific applications such as mathematical reasoning (Yu et al., 2024) and code generation (Zheng et al., 2025), finetuning on supervised data has become a standard practice. A wide range of finetuning methods have been proposed (Hu et al., 2022; Meng et al., 2024; Wang et al., 2025) and systematically analyzed (Biderman et al., 2024; Yang et al., 2024). Despite these advancements, checkpoint selection during multi-epoch finetuning often relies on simple heuristics, such as choosing the final checkpoint (Li et al., 2024; Tong et al., 2024) or selecting based on validation perplexity or accuracy (Huang et al., 2024; Liu et al., 2022a).

In this work, we analyze the learning dynamics of LLM finetuning on reasoning tasks and argue that such heuristic practices may be suboptimal. Specifically, we focus on the study of the **over-memorization** phenomenon, where we define this as a state where the model has excessively memorized training data, leading to high test perplexity while still counter-intuitively maintaining good test accuracy. We further investigate **(1) the conditions inducing** the over-memorization phenomenon and **(2) its detrimental impacts**. Experiments reveal key triggers: higher learning rates accelerate over-memorization onset, while lower learning rates also induce this state given sufficient training epochs, implicating prolonged training duration itself as a fundamental factor regardless of the learning rate. The over-memorization phenomenon is broadly applicable in various tasks, models, as well as finetuning methods like fully finetuning and LoRA-based methods. Although achieving good performance on in-domain benchmarks, over-memorized LLMs tend to exhibit reduced robustness and generalization. Additionally, their abilities to calibrate responses and generate diverse outputs are compromised.

These findings suggest that placing too much emphasis on benchmark accuracy can be risky. It may result in selecting models that rely on memorization rather than demonstrating strong generalization capabilities essential for real-world applications. We propose that the characteristic rise in perplexity during over-memorization reflects the model becoming 'overconfident' and 'stubborn'—over-relying on rigid, memorized pathways instead of flexibly exploring alternatives on new data. Recognizing

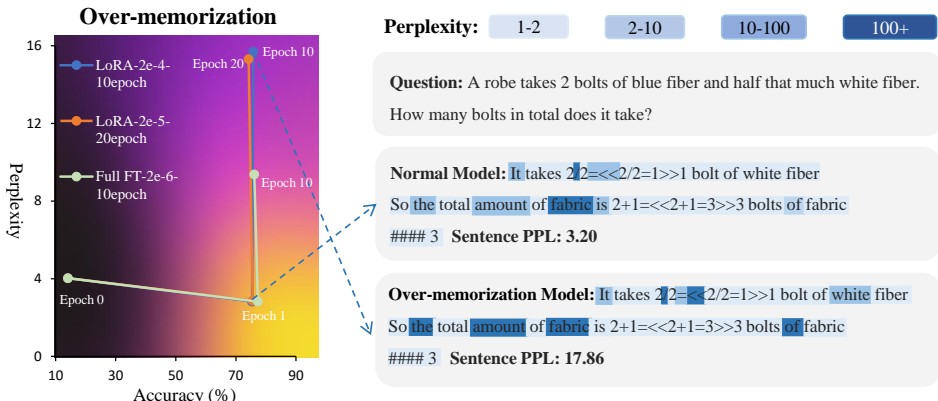

Figure 1: "Over-memorization" in LLM finetuning: a phenomenon where test accuracy remains stable despite increasing test perplexity after extensive training epochs. The left plot (perplexity and accuracy are measured on the GSM8K test set) uses background colors to indicate model states: black (low accuracy), yellow (high accuracy, low perplexity), and purple (high accuracy, high perplexity).

these adverse effects and the phenomenon's dynamics, we identify the problem of the conventional checkpoint selection methods and propose our suggestions based on the observations in our over-memorization experiments. Additionally, we explore two lightweight mitigation strategies, checkpoint merging and memory-aware reweighting loss, which effectively alleviate over-memorization.

## 2 RELATED WORK

Deep neural networks often possess more learnable parameters than training samples, enabling them to simply memorize the data rather than converge to generalizable solutions (Novak et al., 2018). Understanding the phenomenon of memorization in models is therefore crucial for improving generalization performance (Brown et al., 2021; Feldman, 2020; Feldman & Zhang, 2020a). Previous studies have explored various definitions of memorization in deep neural networks (Carlini et al., 2019; 2021a; Feldman & Zhang, 2020b; Zhang et al., 2023). One widely accepted definition is based on differential privacy (Dwork et al., 2006), which formalizes the principle that the removal of any single example from the training set should not substantially alter the resulting model. Based on this definition, some works link stronger memorization during training to reduced generalization ability (Bousquet & Elisseeff, 2000), whereas others emphasize its importance for handling long-tail distributions (Feldman, 2020). In language models, memorization typically refers to generating outputs that resemble specific training examples (Carlini et al., 2021b; Inan et al., 2021; Carlini et al., 2023; Tirumala et al., 2022; Hans et al., 2024). However, some of these works primarily focus on privacy and copyright concerns (Carlini et al., 2021b; Inan et al., 2021; Hans et al., 2024), while others analyze the dynamics of memorization during the pretraining stage (Carlini et al., 2023; Tirumala et al., 2022). The work most similar to ours is Kang et al. (2025), which also investigates memorization during LLM finetuning and specifically analyzes how the finetuned model's generalization behavior is characterized by its pre-memorization training accuracy. In contrast, to the best of our knowledge, we are the first to uncover the phenomenon of over-memorization and investigate its impact on the model's generalization behavior.

## 3 THE OVER-MEMORIZATION PHENOMENON

In classical machine learning, extensive training on limited data often leads to overfitting, where the model performs well on the training set but generalizes poorly to unseen data (Shalev-Shwartz & Ben-David, 2014; Salman & Liu, 2019; Rezaei & Sabokrou, 2023). Overfitted models exhibit an increase in test perplexity and a decrease in test accuracy when the training error decreases or converges (van den Burg & Williams, 2021).

However, our preliminary experiments reveal different training dynamics. We finetune LLaMA-3.1-8B using both LoRA (with learning rates of 2e-4 and 2e-5) and full finetuning (with a learning rate of 2e-6) on the MetaMathQA 10K dataset (Yu et al., 2024). Figure 1(left) presents evaluation results on the GSM8K test set across different finetuning steps. In the early stages of training (e.g., from epoch 0 to epoch 1), the model exhibits a rapid increase in test accuracy and a decrease in perplexity, indicating improved generalization to unseen data. Contrary to the initial rapid gains, a distinct pattern emerges between epochs 1 and 10 for LoRA (lr=2e-5) and full finetuning (lr=2e-6). While test perplexity markedly rises during this period, interestingly, test accuracy resists a corresponding decrease.

Inspired by the aforementioned preliminary experiment observation, we study the **"over-memorization"** phenomenon, where **the model has excessively memorized training data, while still counter-intuitively maintaining good test accuracy**. We focus on the analyses of the over-memorization phenomenon on reasoning tasks like mathematical reasoning. In these tasks, LLMs are finetuned to produce a solution that contains both steps and an answer. While each question is associated with a unique correct answer, the reference output typically illustrates only one possible way to arrive at it. The over-memorized LLMs are likely to generate a reasoning path from the training data and assign high perplexity to other valid paths, while the well-finetuned LLM may generate the correct answer but follow a different reasoning path. Figure 1(right) presents an example that the over-memorized LLM assigns disproportionately low probabilities to alternative tokens or sequences that may still be valid. We further study the over-memorization phenomenon on general tasks in Section 4.4. A brief mechanistic explanation is provided in Appendix C.

By analyzing LLM responses on training and test queries, we can gain insights into the learning dynamics of LLM finetuning with an emphasis on both the correctness of the final prediction and the perplexity of the reference reasoning path. The accuracy measures whether the model produces the correct final answer and directly reflects task performance, solely determined by the final answer. Perplexity (i.e., ppl) is defined as $\mathrm{ppl} = \exp(-\frac{1}{|y^i|} \log(f_\theta(y^i \mid x^i)))$, where $x_i$ is the query input, $y_i$ is the response, $|y^i|$ is the number of tokens in $y_i$. The perplexity on the training set can serve as an indicator of the memorization of the training data. On the test set, perplexity quantifies how probable the model deems this reference path; a higher perplexity indicates that the model assigns lower likelihood to the tokens in the reference solution.

# 4 WHEN DOES THE MODEL OVER-MEMORIZE?

In this section, we investigate the factors that affect the over-memorization of LLMs, such as learning rates, training time, and finetuning methods. We further extend our analyses to different models and tasks, and provide scaling experiments on dataset size and model scale (Appendix E.1). Our findings indicate that higher learning rates cause the model to enter over-memorization earlier, while lower learning rates also lead to over-memorization after more lengthy training. Furthermore, although different finetuning methods exhibit varying levels of adaptation to the learning rates, they all lead to the model's over-memorization. By default, we finetune LLaMA-3.1- 8B on the MetaMathQA 100K dataset and examine their performance on the MetaTrain, MetaTest, and GSM8K dataset (see Appendix B.1 for dataset details) unless otherwise specified.

## 4.1 IMPACT OF LEARNING RATES

We finetune the LLM using the LoRA method across five different learning rates. As shown in Figure 2a, the perplexity on MetaTrain keeps declining, and the model demonstrates varying memorization levels across different learning rates. For instance, a small learning rate (e.g., 2e-6) fails to memorize all data even after 10 epochs, achieving an accuracy below 70%. A learning rate of 2e-4 accelerates memorization compared to 5e-5, suggesting that larger learning rates tend to result in faster learning and memorization of the training data.

The training dynamics of small learning rates (e.g., 2e-5, 2e-6) are similar on MetaTest and GSM8K test as shown in Figure 2b and Figure 2c. In contrast, large learning rates (e.g., 2e-4, 5e-4) exhibit markedly different training dynamics: the perplexity initially declines before subsequently increasing during the training process. The growth is more significant in the GSM8K test set than in MetaTest, as the gap in data distribution is larger between GSM8K and the training set. The perplexity curves

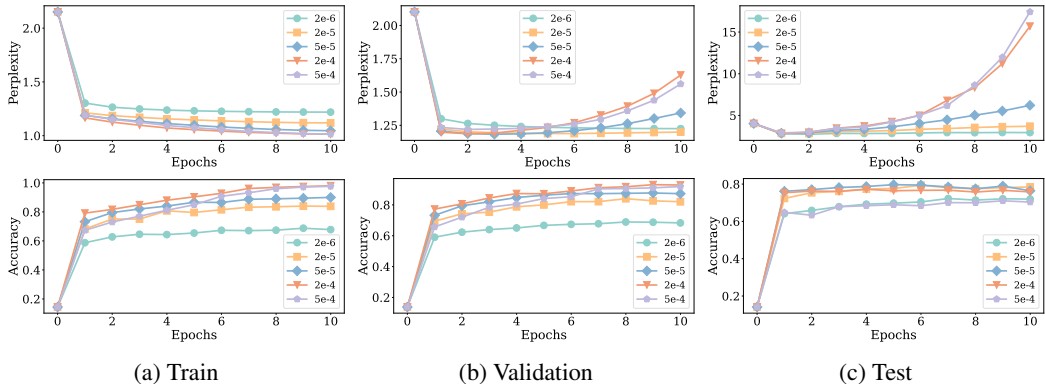

(a) Train        (b) Validation        (c) Test

Figure 2: Results of perplexity and accuracy of LLMs finetuned with different learning rates. The results are reported on MetaTrain (training set), MetaTest (validation set), and GSM8K (test set).

indicate a significant divergence between the predicted and true reasoning trajectories, aligning with our existing understanding of overfitting. Surprisingly, test accuracy remains stable despite a significant increase in test perplexity. This behavior contradicts the common assumption that rising perplexity invariably leads to degraded test accuracy, and we term this specific phenomenon "over-memorization".

From the results in Figure 2b and Figure 2c, we observe that **larger learning rates are more likely to induce over-memorization within the same training epochs**. Although the reference reasoning trajectories have low probabilities, the finetuned LLMs manage to derive the correct answers on the test dataset. The results are hypothesized to be related to the fact that math reasoning tasks often have multiple valid reasoning paths. The over-memorizing LLM still retains the possibility of exploring alternative valid reasoning paths.

## 4.2 IMPACT OF TRAINING EPOCHS

Small learning rates like 2e-5 result in relatively low test perplexity even after training for 10 epochs, as shown in Figure 2c. To further investigate whether smaller learning rates also exhibit over-memorization under lengthy training, we finetuned the LLM with a learning rate of 2e-5 for 20 epochs. As shown in Figure 1, the perplexity on the test set increased to 16, which is similar to 2e-4, indicating a substantial divergence between model predictions and reference answers. However, the test accuracy remained above 70%, demonstrating that even with a small learning rate, prolonged training can result in over-memorization. These findings highlight that while smaller learning rates may delay the onset of over-memorization, they are not immune to this phenomenon when models are trained for an excessively long time. This reinforces the importance of carefully monitoring both perplexity and accuracy during training to prevent over-memorization, regardless of the learning rate.

## 4.3 IMPACT OF FINETUNING METHODS

In this section, we study over-memorization under four parameter-efficient finetuning and one full finetuning method (Hu et al., 2022; Meng et al., 2024; Wang et al., 2025). The specific experimental configurations are detailed in Table 6.

The results are shown in Figure 3. Our experiments reveal that **over-memorization occurs not only in LoRA but also across various finetuning methods**. Specifically, under different finetuning methods, larger learning rates are more likely to lead to over-memorization. Fully finetuning is observed to exhibit higher perplexity compared to LoRA-based methods. Notably, over-memorization in full finetuning not only emerges in lengthy training. As shown in Figure 3(a)-(b), the test perplexity in full finetuning already begins to increase around epoch 3, earlier than in other methods. This suggests that full finetuning may be more prone to early over-memorization, which becomes more prominent as training continues. We attribute this to the fact that fully finetuning updates a larger number of parameters. To further validate this assumption, we design LoRA+, a variant of LoRA.

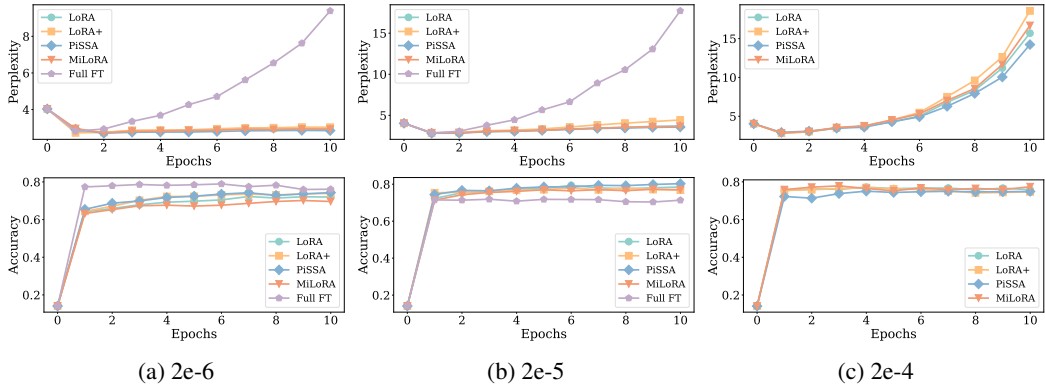

Figure 3: Results of different finetuning methods at three different learning rates, assessing both test perplexity and accuracy.

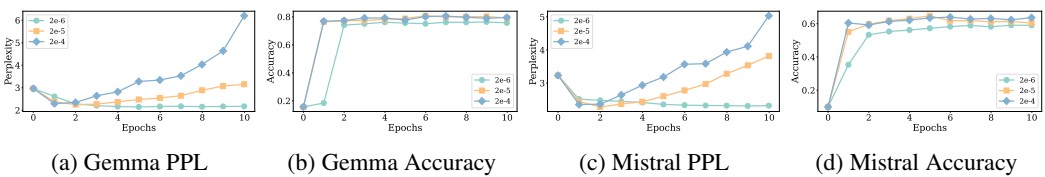

Figure 4: Test perplexity and accuracy curves illustrating over-memorization in Gemma-2-9B and Mistral-7B model.

The only difference is that the attention output linear projector and feed-forward gate weight are also finetuned. Under the same settings, LoRA+ exhibits higher perplexity than LoRA, confirming that **increasing the number of finetuned parameters leads to higher perplexity**. More comprehensive results across a wider range of learning rates, including training, validation, and test performance, are provided in Appendix E.

## 4.4 OVER-MEMORIZATION ON DIVERSE TASKS AND DIVERSE MODELS

**Diverse Task**   To examine the broader applicability of the over-memorization phenomenon beyond mathematical reasoning, we broaden our study to include code generation, scientific question answering, and open-ended text generation (Appendix E). Across all tasks, we finetune LLaMA-3.1-8B using LoRA (details in Appendix Table 6). For code generation, models are trained on CodeFeedback 10K (Zheng et al., 2025) and evaluated on HumanEval (Chen et al., 2021a). Scientific QA uses the GPQA benchmark (Rein et al., 2024), with 300 training examples and the remainder for evaluation, leveraging gold reasoning chains for perplexity computation.

Table 1: Over-memorization across diverse tasks.

| Task | Metrics | Epoch 0 | Epoch 2 | Epoch 4 | Epoch 6 | Epoch 8 | Epoch 10 |
|---|---|---|---|---|---|---|---|
| GPQA | PPL | 4.80 | 4.51 | 7.47 | 14.77 | 30.08 | 40.53 |
| | Accuracy (%) | 5.41 | 21.62 | 22.30 | 25.00 | 22.97 | 25.00 |
| HumanEval | PPL | 1.66 | 1.82 | 2.13 | 2.58 | 3.26 | 4.08 |
| | Accuracy (%) | 37.80 | 46.95 | 48.17 | 50.61 | 48.17 | 50.61 |

As illustrated in Table 1, our findings consistently indicate the presence of over-memorization across a diverse set of domains. A recurring pattern emerges: test perplexity increases rapidly during finetuning, while task-specific evaluation metrics remain high. For instance, in HumanEval, finetuning with a learning rate of 2e-4 led to a sharp rise in test perplexity, despite test accuracy remaining stable, aside from minor fluctuations attributable to the small size of the evaluation set (164 samples). Similar trends appear in GPQA, where perplexity steadily increases even as final answer accuracy remains largely unaffected. These results provide compelling evidence that over-memorization is a pervasive and task-agnostic phenomenon.

Table 2: Accuracy of normal and over-memorized models on ID and OOD mathematical reasoning benchmarks. The Avg. is the average result over OOD test sets.

| Methods | ID Test Set | | OOD Test Set | | | | | | |
|---|---|---|---|---|---|---|---|---|---|
| | GSM8K | MATH | SVAMP | ASDiv | MAWPS | TabMWP | Minerva | MMLU_STEM | Avg. |
| Normal model | 75.6 | 28.9 | 77.1 | 81.7 | 89.7 | 67.1 | 27.4 | 17.9 | 60.2 |
| Over-memorized model | 76.4 | 28.5 | 75.3 | 79.1 | 89.6 | 63.6 | 29.2 | 12.0 | 58.1 |

**Diverse Model**    To further evaluate the generality of the over-memorization phenomenon across different model architectures, we expand our analysis to Mistral-7B-v0.3(Jiang et al., 2023) and Gemma-2-9B(Team, 2024a). Both models are finetuned on the MetaMath-10K dataset using LoRA. As shown in Figure 4, both models exhibit clear signs of over-memorization, manifested as rising test perplexity despite sustained or even improved test accuracy. These results are consistent with our earlier observations and further reinforce the finding from Section 4.1: under the same model, larger learning rates tend to induce a more rapid increase in test perplexity, thereby leading to over-memorization more quickly.

## 5 BEHAVIORAL ANALYSES

Over-memorized models often achieve high test accuracy and seem effective. However, these models typically exhibit higher perplexity on the test set, raising the question of whether they are truly performing well. To address this, we evaluate the model from multiple perspectives, including robustness, OOD performance, diversity, privacy risk, and calibration. To analyze this, we select the LoRA checkpoints from epoch 3 and epoch 10 with a learning rate of 2e-4, representing the normal model and the over-memorized model, respectively. Both models achieve similar performance on the GSM8K test set (75.6 vs. 76.4 for the normal and over-memorized model, respectively), while the test perplexity of the over-memorized model exceeds that of the normal model by 4.52 times. Please refer to Appendix D.1 and Appendix D.2 for details of the evaluations on model calibration and privacy risk.

### 5.1 PERFORMANCE ON OOD DATASET

In Section 4, we previously observed that the over-memorized model generalizes well on in-distribution (ID) test sets. A natural question that arises is how it performs on out-of-distribution (OOD) test sets. In this section, we explore this by comparing normal and over-memorized models on the OOD mathematical benchmarks, including SVAMP (Patel et al., 2021), ASDiv (Miao et al., 2020), MAWPS (Koncel-Kedziorski et al., 2016), TabMWP (Lu et al., 2023), Minerva_MATH (Lewkowycz et al., 2022), and MMLU_STEM (Hendrycks et al., 2020). We also include results on ID test sets for reference. In addition to GSM8K discussed in Section 3, we add results on MATH (Hendrycks et al., 2021). Since MATH is one of the seed datasets used to synthesize MetaMathQA, it is also considered an ID test set.

As shown in Table 2, the over-memorized model performs comparably to the normal model on in-domain (ID) test sets, but falls behind by an average of 2.1 points on out-of-domain (OOD) benchmarks. This suggests that over-memorization does not harm ID generalization, but undermines robustness to distribution shifts. The effect becomes more pronounced as the gap between training and test distributions increases, indicating that over-memorized models are particularly vulnerable to OOD evaluation. In summary, over-memorized models suffer from generalization problems and have poor performance on OOD data, like overfitting models. However, the generalization defects of overfitting models are more pronounced, as they exhibit poor performance even on ID samples.

### 5.2 ROBUSTNESS AGAINST PROMPT PERTURBATION

We begin by comparing the normal and over-memorized models on their robustness against prompt perturbation. Specifically, for each test sample, we combine the original training prompt (see Appendix B.3) and the test input as the query. Then, we present the model with the original queries and also with queries where short, neutral preambles are appended (Kang et al., 2025). These

preambles are designed to be plausible introductory phrases that a model might itself generate, without affecting the core problem or its answer. The specific preambles are:

- **First**: First, let's tackle this step by step together.
- **Today**: Today, we'll work through this problem together and find a clear solution.
- **We**: We understand the problem we're tackling and will work together to solve it.
- **Good**: That's a great question! We understand the problem we're tackling and will work together to solve it.

We examine the model's behavior under slight variations in the input prompt. Given that these neutral additions resemble typical openings in valid reasoning, a robust model should still produce the correct solution. Conversely, if it fails, the model likely only regurgitates the training response. Table 3 shows the evaluation results on the GSM8K test set. While the over-memorized model performs better than the normal model

Table 3: Accuracy of the normal and over-memorized model with and without prompt perturbation on the GSM8K dataset.

| Models | w.o Perturb | First | Today | We | Good | Avg. |
|---|---|---|---|---|---|---|
| Normal | 75.6 | 76.7 | 72.0 | 73.9 | 73.6 | 74.1 |
| Over-memorized | 76.4 | 74.7 | 70.3 | 72.0 | 72.5 | 72.4 |

on unaltered prompts, its performance significantly degrades with perturbed prompts, averaging 1.7 points lower than the normal model. These findings suggest that **over-memorization negatively impacts the model's robustness**.

### 5.3 Diversity of Generations

Inference-time techniques that generate multiple outputs (Welleck et al., 2024; Snell et al., 2025), such as Best-of-N (BoN) sampling (Charniak & Johnson, 2005; Stiennon et al., 2020), can enhance LLM reasoning. The effectiveness of these methods often correlates with output diversity, as a wider exploration of the solution space typically yields better results (Chow et al., 2025). To this end, we compare these model types by generating $N$ outputs per query (Figure 5) and evaluate them using two metrics: BoN accuracy,

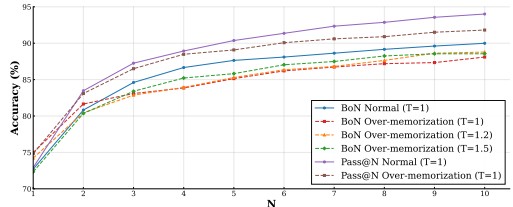

Figure 5: BoN and PassN accuracy of normal models and over-memorized models on the GSM8K.

where the optimal sample from $N$ candidates is identified by the Skywork-o1-Open-PRM-Qwen-2.5-7B process reward model (o1 Team, 2024), and Pass@N accuracy, which measures if at least one correct answer is present among the $N$ samples (Chen et al., 2021b).

The results in Figure 5, using a sampling temperature of 1.0, demonstrate that the conventionally trained model consistently surpasses the over-memorized model in both BoN and Pass@N accuracy for $N \geq 3$. To ascertain whether this disparity might be an artifact of an inadequately low temperature for the over-memorized model, its BoN performance was further assessed across temperatures from 1.0 to 2.0. This analysis confirmed that its performance persistently lagged behind the conventionally trained model, especially for larger $N$. These findings suggest that **over-memorized models tend to generate less diverse outputs**, thereby diminishing the performance gains achievable through repeated sampling techniques for reasoning tasks. This trend is further corroborated by additional lexical and semantic diversity metrics reported in Appendix D.3.

## 6 Methods for Mitigating Over-Memorization

### 6.1 Receipt for Checkpoint Selection

When finetuning large models, previous work either finetunes the model for 1-4 epochs and selects the last checkpoint as the final model (Meng et al., 2024; Wang et al., 2025), or alternatively, performs model selection or early stopping based on the perplexity or accuracy on a validation set (Zhang et al., 2024a; Huang et al., 2024; Lin et al., 2024). However, we find that due to the over-memorization effect, solely relying on validation perplexity or accuracy can lead to suboptimal results.

To verify this, we finetune the model using LoRA (Hu et al., 2022) and PiSSA (Meng et al., 2024) with a learning rate of 2e-4, employing MetaTest as the validation set, GSM8K, and MATH as the ID test

sets, and the same OOD test sets as in Section 5.1. As shown in Table 4, we report the average performance on both ID and OOD test sets using different model selection metrics. We observe that validation perplexity tends to select an earlier checkpoint, whereas validation accuracy tends to select a later checkpoint. This trend is consistent across various finetuning methods and learning rate combinations.

In these cases, validation accuracy gradually increases, while validation perplexity initially decreases and then increases as training progresses. However, when using validation perplexity for model selection, the final model's ID performance is suboptimal, with gaps of 1.30 and 2.95 compared to the best ID accuracy achieved for LoRA and PiSSA, respectively. This also indicates that early stopping based on an increase in validation perplexity is not ideal, as continued training can further enhance the model's performance on the ID test sets. In contrast, selecting models based on validation accuracy results in good ID performance. Nevertheless, these models tend to perform poorly on OOD data and exhibit various issues discussed in Section 5, such as robustness, calibration, diversity, and privacy concerns.

Table 4: Model selection results with different selection metrics. We finetune the model utilizing LoRA and PiSSA with a learning rate of 2e-4 and MetaTest as the validation set. ACC denotes accuracy. Selection with ID and OOD accuracy is reported for reference.

| FT Method | Selection Metric | Position | ID ACC | OOD ACC |
|---|---|---|---|---|
| LoRA | ID ACC | epoch 9 | **53.00** | 67.88 |
| | OOD ACC | epoch 1 | 51.25 | **71.32** |
| | Valid PPL | epoch 2 | 51.70 | 70.56 |
| | Valid ACC | epoch 9 | **53.00** | 67.88 |
| PiSSA | ID ACC | epoch 10 | **51.40** | 65.52 |
| | OOD ACC | epoch 1 | 48.55 | **68.04** |
| | Valid PPL | epoch 2 | 48.45 | 65.46 |
| | Valid ACC | epoch 10 | **51.40** | 65.52 |

To address the above issues, we offer the following suggestions: (1) When finetuning LLMs, it is advisable to use a small number of epochs, typically between 1 and 4. This reduces the likelihood of over-memorization, allowing the last checkpoint to be used as the final model. (2) For model selection, it is recommended to use a combination of validation accuracy and perplexity, choosing models with the highest validation accuracy within a certain range of validation perplexity.

## 6.2 MITIGATING OVER-MEMORIZATION VIA CHECKPOINT MERGING

Our first strategy mitigates over-memorization by merging checkpoints from different training stages. Earlier checkpoints (e.g., epoch 3) tend to preserve higher output diversity and OOD performance, whereas later checkpoints (e.g., epoch 10) achieve stronger ID accuracy but suffer from reduced diversity and OOD accuracy, as discussed in Section 5. By merging them, we aim to combine these complementary strengths, effectively ensembling the capabilities of different training stages (Dang et al., 2025; Wortsman et al., 2022).

Concretely, we follow the training and evaluation setup in Section 5, where we finetune LLaMA-3.1-8B with LoRA (learning rate $2 \times 10^{-4}$) on MetaMath-100K and subsequently evaluate the model's ID/OOD performance, as well as its generation diversity. Let $\theta_{\text{base}}$ denote the pre-trained parameters, $\theta_3$ the parameters after epoch 3, and $\theta_{10}$ those after epoch 10. We compute deltas $\Delta_3 = \theta_3 - \theta_{\text{base}}$ and $\Delta_{10} = \theta_{10} - \theta_{\text{base}}$, and define the merged model as:

$$\theta_{\text{merge}} = \theta_{\text{base}} + \tfrac{1}{2}(\Delta_3 + \Delta_{10}). \tag{1}$$

As shown in Figure 6a and Table 5, checkpoint merging gets the best Pass@1 across all checkpoints and also has a great Pass@10 accuracy. Additionally, SFT Merge achieves a great average OOD accuracy, surpassing epoch 3 and epoch 10 by 1.2 and 3.3 points, respectively. These results indicate that merging effectively alleviates over-memorization without requiring additional training.

## 6.3 MITIGATING OVER-MEMORIZATION VIA MEMORIZATION-AWARE REWEIGHTING

While merging provides a training-free solution, we further mitigate over-memorization by modifying the finetuning objective itself. Standard SFT assigns equal weight to all tokens, regardless of their learning difficulty. As a result, frequently memorized tokens dominate optimization, reinforcing rigid memorization rather than improving generalization.

We introduce **Memorization-Aware Reweighting (MAR)**, which downweights tokens that the model already predicts with high confidence and upweights those with lower probabilities. Formally, for

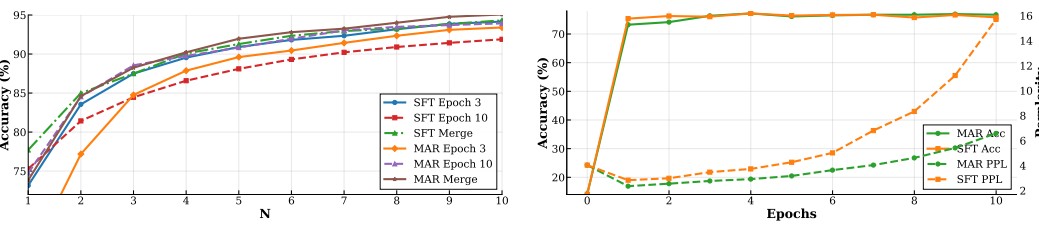

(a) Pass@N comparison across methods.  (b) Accuracy and perplexity curves during training.

Figure 6: Evaluation of over-memorization mitigation methods. (a) Pass@N comparison across methods. (b) Accuracy (left) and perplexity (right) during 10 training epochs.

Table 5: Accuracy of methods on in-domain (ID) and out-of-domain (OOD) benchmarks. Avg. denotes the average over OOD test sets. Detailed descriptions of OOD benchmarks are in Section 5.1.

| Methods | ID Test Set | | OOD Test Set | | | | | | |
| | GSM8K | MATH | SVAMP | ASDiv | MAWPS | TabMWP | Minerva | MMLU_STEM | Avg. |
|---|---|---|---|---|---|---|---|---|---|
| SFT Epoch 3 | 75.6 | 28.9 | 77.1 | 81.7 | 89.7 | 67.1 | 27.4 | 17.9 | 60.2 |
| SFT Epoch 10 | 76.4 | 28.5 | 75.3 | 79.1 | 89.6 | 63.6 | 29.2 | 12.0 | 58.1 |
| SFT Merge | 78.5 | 31.4 | 77.8 | 81.5 | 90.4 | 68.9 | 31.6 | 17.9 | 61.4 |
| MAR Epoch 3 | 76.7 | 29.2 | 74.4 | 81.2 | 90.1 | 68.9 | 29.2 | 17.1 | 60.2 |
| MAR Epoch 10 | 76.4 | 29.6 | 74.5 | 80.8 | 90.2 | 68.7 | 29.6 | 21.4 | 60.9 |
| MAR Merge | 77.1 | 31.2 | 76.7 | 82.0 | 91.4 | 68.4 | 33.2 | 23.5 | 62.5 |

each token $y_t$, we compute its predicted probability $p_\theta(y_t \mid x, y_{<t})$ and reweight its contribution by $1 - p_\theta(y_t \mid x, y_{<t})$:

$$\mathcal{L}_{\text{MAR}} = \sum_{t=1}^{T} \left(1 - p_\theta(y_t \mid x, y_{<t})\right) \cdot \ell(y_t), \tag{2}$$

where $\ell(y_t)$ is the standard cross-entropy loss. This adjustment encourages the model to focus on tokens that remain uncertain, reducing excessive memorization of already-learned patterns.

Figure 6b and Table 5 summarize the results. Compared with SFT, MAR maintains comparable performance across both ID and OOD benchmarks but achieves substantially lower perplexity throughout training, with less than half of SFT's perplexity at epoch 10. On OOD benchmarks, MAR does not suffer from performance degradation due to additional training; in fact, its accuracy at epoch 10 even surpasses that of MAR at epoch 3 and SFT at the same epoch. Furthermore, unlike SFT, MAR's Pass@N accuracy at epoch 10 exceeds that at epoch 3, further suggesting that additional training does not lead to over-memorization in MAR. In addition, merging MAR checkpoints from different training stages (Epoch 3 and Epoch 10) further improves performance, yielding the highest Pass@10 accuracy and strongest OOD results among all evaluated methods. These results demonstrate that our proposed methods can effectively mitigate over-memorization, which achieves better ID accuracy compared to SFT, while preserving strong output diversity and OOD performance (see MAR Epoch 10 and MAR Merge).

# 7 CONCLUSION

In this paper, we introduce and systematically analyze the phenomenon of over-memorization in LLM finetuning, where test accuracy remains stable despite a rapid increase in test perplexity. We study the factors such as learning rates, training epochs, and fine-tuning methods that influence over-memorization. We analyze its impact on LLM model behavior, including negative effects on model robustness, OOD performance, calibration, generation diversity, and privacy protection. Our scaling analyses show that over-memorization is common regardless of the finetuning dataset scale or the LLM model size. Our findings enhance practitioners' understanding of LLM finetuning, providing insights into checkpoint selection and introducing methods to mitigate over-memorization. Additionally, our research highlights that overparameterized pretrained LLMs possess unique properties distinct from traditional machine learning models. We encourage future research to further empirically and theoretically investigate the training and generalization mechanisms of LLMs.

ETHICS STATEMENT

Our study introduces the "over-memorization" phenomenon in LLM finetuning, which is particularly relevant for ensuring the reliability and trustworthiness of finetuned LLMs. This work highlights that accuracy alone is an insufficient measure of a model's true capabilities. Future efforts should therefore encourage more nuanced analysis and comprehensive evaluation of LLMs, fostering the development of responsible, helpful, and trustworthy AI systems for diverse real-world applications.

REPRODUCIBILITY STATEMENT

This work primarily presents an analysis of the "over-memorization" phenomenon in LLM fine-tuning. Full details of the training setup, including dataset configuration, hyperparameters for each fine-tuning method, and prompt designs, can be found in Appendix B of the paper.

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

## LIMITATION

While our work introduces the over-memorization phenomenon in LLM finetuning, we acknowledge that our work has a limitation in that our analysis predominantly focused on reasoning tasks. Although supplementary AlpacaEval experiments were included, the exploration of this phenomenon in open-ended generation tasks remains less comprehensive and warrants further study.

## A    LLM USAGE

In the preparation of this paper, we only used large language models (LLMs) as an assistive tool for grammar correction and text polishing.

## B    EXPERIMENT SETUP

This section presents the detailed experimental configuration, including the finetuning parameters, datasets used for training and evaluation, and the prompt format.

### B.1    DATASET SETUP

Unless otherwise specified, this paper utilizes LLaMA-3.1-8B as the base language model, with the first 100K samples from the MetaMathQA dataset serving as the training set. A subset of this data, denoted as MetaTrain, is first sampled to monitor training dynamics such as perplexity and accuracy. An additional set, MetaTest, is constructed from the remaining, unseen portion of MetaMathQA. Both MetaTrain and MetaTest are set to have the same sample size as the GSM8K test set. Since

MetaTest originates from the same distribution as the training data, it is inherently more similar to the training set than GSM8K. Therefore, for clarity and ease of distinction, we designate MetaTest as the validation set and GSM8K as the test set by default. MetaMathQA is derived from GSM8K and MATH, making all three datasets share similar formats and reasoning styles. As a result, in Section 5.1, we treat GSM8K and MATH as in-distribution (ID) test sets.

## B.2 HYPERPARAMETER CONFIGURATION

Table 6: Hyperparameters for different finetuning methods.

|  | Full Finetuning | LoRA | MiLoRA | PiSSA | LoRA+ |
|---|---|---|---|---|---|
| Batch Size | 128 | 128 | 128 | 128 | 128 |
| Optimizer | AdamW | AdamW | AdamW | AdamW | AdamW |
| LR Scheduler | Linear | Linear | Linear | Linear | Linear |
| Warmup Step | 100 | 100 | 100 | 100 | 100 |
| Epoch | 10 | 10 | 10 | 10 | 10 |
| LoRA Dropout | - | 0.05 | 0.05 | 0.05 | 0.05 |
| LoRA Rank | - | 64 | 64 | 64 | 64 |
| LoRA Alpha | - | 128 | 64 | 64 | 128 |
| Target | - | $q_{\mathrm{proj}},k_{\mathrm{proj}},v_{\mathrm{proj}},\mathrm{up}_{\mathrm{proj}},\mathrm{down}_{\mathrm{proj}}$ | | | Add $o_{\mathrm{proj}},\mathrm{gate}_{\mathrm{proj}}$ |

Table 6 summarizes the hyperparameter settings used for different finetuning strategies. All experiments are conducted on NVIDIA A100-80G GPUs, using a batch size of 128 to ensure training stability. These settings are inspired by the LLM-Adapter configuration in Hu et al. (2023), with some adjustments to suit our setup. As a reference point, finetuning with LoRA on the MetaMath-10K subset takes approximately 1.5 hours on two NVIDIA A100-80G GPUs.

## B.3 PROMPT FORMAT

The prompt format used for both training and evaluation follows a standard instruction-following template for the mathematics task. The prompt is shown below:

> **Training and Evaluation Prompt**
>
> Below is an instruction that describes a task. Write a response that appropriately completes the request.
>
> ### Instruction:
> {instruction}
>
> ### Response:

## C MECHANISTIC EXPLANATION OF OVER-MEMORIZATION

The paradoxical coexistence of high accuracy and high perplexity in over-memorized LLMs can be explained by the cross-entropy (CE) loss used in supervised finetuning. Given input $x$ and target sequence $y_{1:T}$,

$$\mathcal{L}_{\mathrm{CE}}(\theta) = -\sum_{t=1}^{T} \log p_\theta(y_t \mid x, y_{<t}), \tag{3}$$

with gradient

$$\frac{\partial \mathcal{L}_{\mathrm{CE}}}{\partial z_t(j)} = p_t(j) - \mathbf{1}\{j = y_t\}. \tag{4}$$

Here, the annotated token $y_t$ is consistently reinforced, while all alternatives $a \neq y_t$ are uniformly suppressed. Because $p_t(a) \propto e^{z_t(a)}$, repeated updates drive an exponential decay in their probabilities. The dominant reasoning path is preserved, but alternative paths become increasingly implausible.

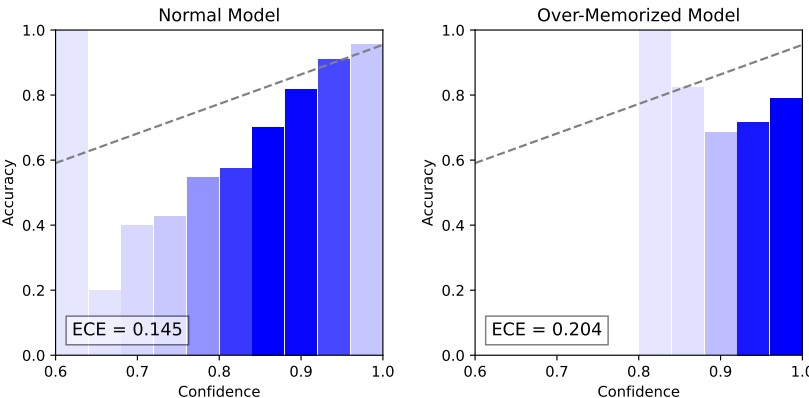

Figure 7: Calibration of the normal model and the over-memorized model.

This mechanism explains why the set of high-perplexity tokens remains stable between a normally finetuned model and an over-memorized model, yet their perplexity magnitudes escalate dramatically. Accuracy remains unaffected because the dominant path is still valid, while perplexity rises due to systematic discouragement of all other continuations.

# D   ADDITIONAL BEHAVIORAL ANALYSES OF OVER-MEMORIZED MODELS

The main paper (Section 5) discusses various behavioral aspects of over-memorized models. Due to space constraints in the main text, the detailed evaluations of model calibration (Appendix D.1) and membership inference attack performance (Appendix D.2) are presented in this section. These analyses provide further insights into the characteristics of over-memorized models.

## D.1   MODEL CALIBRATION

Calibration requires that the probabilities assigned by the model to its predictions (i.e., confidence) match the actual correctness of those predictions (i.e., accuracy) (Wang et al., 2020). We use Expected Calibration Error (ECE) to measure the calibration of the model, which is a commonly used metric for evaluating calibration error. ECE measures the expected difference between confidence and accuracy (Naeini et al., 2015). Specifically, ECE divides the predictions into $M$ bins $\{B_1, B_2, \ldots, B_M\}$ based on their confidence and computes the weighted average of the accuracy-confidence difference for each bin:

$$ECE = \sum_{m=1}^{M} \frac{|B_m|}{N} \left| \mathrm{acc}(B_m) - \mathrm{conf}(B_m) \right| \tag{5}$$

where $N$ is the number of prediction samples and $|B_m|$ is the number of samples in the $m$-th bin.

Figure 7 presents a comparative analysis of calibration between the normal model and the over-memorized model. For this analysis, predictions are generated using a sampling configuration with a temperature of $1.0$ and top_p of $1.0$; this setup is chosen to foster diverse yet plausible outputs. The confidence of each prediction is determined by calculating the average token probability across the entire generated sequence. As depicted in the figure, the color intensity within the bars indicates the density of samples per confidence interval, with darker shades representing higher concentrations of samples.

The normal model exhibits superior calibration, achieving an ECE of $0.145$, which is lower than the $0.204$ ECE of the over-memorized model. This difference suggests that the normal model provides more reliable confidence estimates, thereby enabling users to better assess the likely correctness of its predictions. In contrast, the **over-memorized model displays a marked tendency to overestimate its confidence.** This is evidenced by the majority of its predictions registering confidence levels above $0.9$. Such overconfidence makes it more challenging for users to accurately discern the reliability of the model's outputs.

Table 7: Performance of normal models and over-memorized models under different membership inference attack methods.

| | Loss | | Zlib | | Min-K% | | Min-K++% | |
|---|---|---|---|---|---|---|---|---|
| | AUROC | TPR@5%FPR | AUROC | TPR@5%FPR | AUROC | TPR@5%FPR | AUROC | TPR@5%FPR |
| Normal Model | **58.5** | **12.0** | **55.0** | **13.6** | **58.5** | **12.4** | **65.0** | **14.2** |
| Over-memorized Model | 72.2 | 22.4 | 65.5 | 21.2 | 72.2 | 22.8 | 75.5 | 26.4 |

Table 8: Measuring the impact of over-memorization on generation diversity. EAD and Sentence-BERT are used as diversity metrics, where higher values indicate greater diversity.

| Base Model | Finetuning Methods | Over-memorization | EAD | Sentence-BERT |
|---|---|---|---|---|
| Llama3.1-8B | LoRA | Normal Model | 47.71 | 9.08 |
| | | Over-memorized Model | 37.71 | 6.57 |
| | Full finetuning | Normal Model | 48.66 | 10.40 |
| | | Over-memorized Model | 35.52 | 6.40 |
| Mistral-7B-v0.3 | LoRA | Normal Model | 51.62 | 11.59 |
| | | Over-memorized Model | 34.61 | 6.32 |
| | Full finetuning | Normal Model | 54.20 | 15.50 |
| | | Over-memorized Model | 37.39 | 8.39 |
| Gemma-2-9B | LoRA | Normal Model | 48.09 | 9.19 |
| | | Over-memorized Model | 31.04 | 4.51 |
| | Full finetuning | Normal Model | 48.62 | 10.46 |
| | | Over-memorized Model | 30.77 | 5.27 |

## D.2 MEMBERSHIP INFERENCE ATTACK PERFORMANCE

Membership Inference Attacks (MIA), which aim to determine whether a given data instance has been used for model finetuning, can uncover underlying privacy concerns associated with the model. Previous work (Yu et al., 2022; Fu et al., 2024) has identified finetuning as the stage most susceptible to privacy leaks, due to the relatively small and often private datasets used in this process. Therefore, this section compares the MIA attack performance of two finetuned LLMs, namely normal and over-memorized models, to understand their susceptibility to privacy leaks.

Specifically, we construct a test dataset with two sets of records: a member set, which contains 500 samples from the training set, and a nonmember set, which includes 500 unseen samples from the held-out MetaMathQA dataset. Given a finetuned LLM, MIA predicts whether each test sample belongs to the member or nonmember set. Since MIA is essentially a binary classification task, we follow previous work (Zhang et al., 2024b) and report the AUROC and TPR@5%FPR as the attack metrics, where higher scores indicate more severe privacy leakage. As presented in Table 7, we compare normal and over-memorized models using four widely used MIA methods: Loss (Yeom et al., 2018), Zlib (Carlini et al., 2021b), Min-k% (Shi et al., 2023), and Min-k%++ (Zhang et al., 2024b). As can be seen, the AUROC of the normal model across all MIA methods remains in the range of 55%-65%, suggesting a moderate risk of privacy leakage. In contrast, the over-memorized model exhibits a notable increase in AUROC, ranging from 70%-75%. This trend is further supported by the TPR@5%FPR metric. Consequently, **over-memorized models demonstrate greater susceptibility to membership inference attacks**, underscoring the heightened privacy risks associated with excessive memorization.

## D.3 MORE DIVERSITY METRIC

Although the BoN and Pass@N results reported in the Section 5.3 provide some insight into the diversity of generated outputs, we further supplement this analysis with additional diversity metrics in this section: **EAD** (Expectation-Adjusted Distinct N-grams) and **Sentence-BERT diversity**. EAD measures lexical diversity by counting distinct 1- to 5-grams, and includes an adjustment mechanism to mitigate bias from shorter outputs (Li et al., 2016; Liu et al., 2022b). Sentence-BERT diversity measures semantic diversity, calculated as $1 -$ average cosine similarity between Sentence-BERT embeddings of generated responses (Kirk et al., 2024).

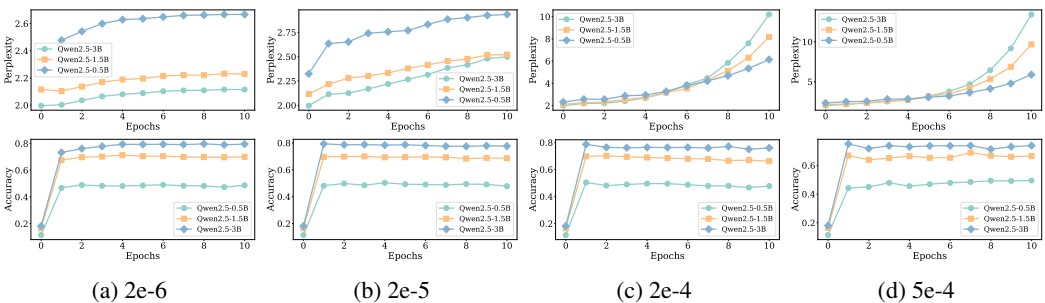

Figure 8: Evaluating the Qwen2.5 model of different sizes when finetuned with four different learning rates, assessing both test perplexity and accuracy.

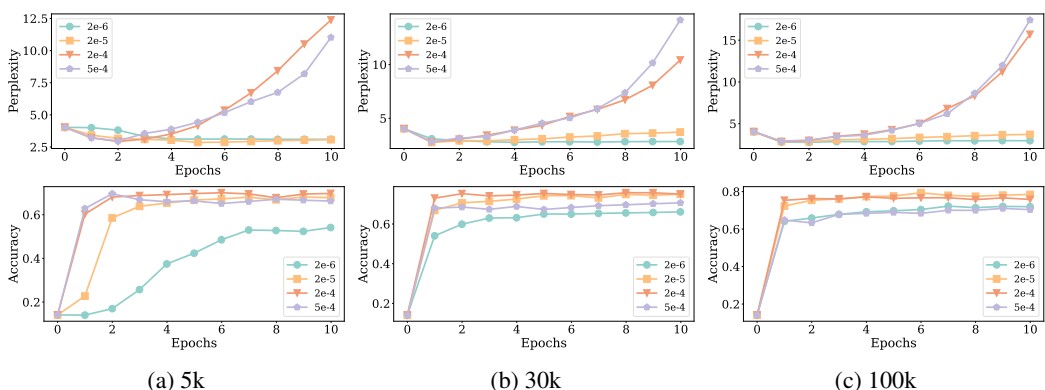

Figure 9: Evaluating LLM finetuned with varying size of the dataset, assessing both test perplexity and accuracy.

In addition to LLaMA3.1-8B, we also include results for Gemma-2-9B and Mistral-7B-v0.3, under both LoRA and full finetuning settings. As shown in Table 8, over-memorized models consistently demonstrate substantially lower diversity compared to their normally finetuned models. These findings confirm that over-memorization reduces the diversity of model generations, thereby limiting their effectiveness in sampling-based inference-time strategies.

# E   SUPPLEMENTARY EXPERIMENTAL RESULTS

## E.1   SCALING ANALYSIS

In the previous section, we demonstrated that over-memorization occurs broadly across tasks and architectures. Here, we deepen our analysis by investigating how two key scaling dimensions—training dataset size and model size—affect the over-memorization phenomenon.

**Effect of Model Size**   We further investigate the effect of model scale on the over-memorization phenomenon using Qwen2.5 models (Team, 2024b) of varying sizes (0.5B, 1.5B, and 3B), with all finetuning performed via LoRA, consistent with our main experimental setup. Figure 8 reveals several key trends: (1) Larger models generally yield better test performance. (2) Over-memorization is more pronounced in larger models, as indicated by substantially higher test perplexity in the overtraining regime (Figures 8(c)–(d)). (3) Larger models exhibit stronger initial generalization, achieving lower test perplexity early in training. However, their greater capacity for memorization (Tirumala et al., 2022) leads to a faster increase in test perplexity over time, making them more susceptible to over-memorization (Figures 8(c)–(d)). (4) Larger models favor smaller optimal learning rates, as shown in

Figure 15 (Appendix E). This may be due to the aggregate effect of conservative updates across a large number of parameters, enabling effective optimization even with small per-parameter steps.

**Effect of Data Size** To assess the effect of data size, we finetuned Llama-3.1-8B on subsets of the MetaMath dataset of varying sizes: 5K, 30K, and 100K samples. For each dataset size, we explored multiple learning rates to identify optimal configurations and observe performance trends.

The results in Figure 9 reveal several key insights: (1) The over-memorization pattern—characterized by rising test perplexity despite stable accuracy—persists across all dataset sizes. (2) In line with prior work (Zhang et al., 2024a), larger datasets generally yield better test performance. (3) Most notably, we observe a systematic interaction between dataset size and optimal learning rate: **larger datasets favor smaller learning rates**. For instance, the optimal learning rate shifts from 2e-4 on the 5K subset to 2e-5 on the 100K subset, with the 30K subset performing well under both. This behavior likely stems from the increased number of optimization steps induced by larger datasets under fixed training epochs and batch size, making smaller learning rates sufficient to traverse the solution space.

### E.2 ADDITIONAL OPEN-ENDED TASK

We investigate whether similar dynamics appear in open-ended generation tasks, where evaluation typically considers the entire generated output rather than a single final answer. We conduct experiments using AlpacaEval 2.0 (Dubois et al., 2024), finetuning with LoRA on 10K samples

Table 9: PPL and LCWR across different epochs

| Metrics | Epoch 0 | Epoch 1 | Epoch 2 | Epoch 3 | Epoch 6 | Epoch 8 | Epoch 10 |
|---------|---------|---------|---------|---------|---------|---------|----------|
| PPL | 4.67 | 2.90 | 2.95 | 3.23 | 6.67 | 12.65 | 21.88 |
| LCWR | 0.21 | 5.93 | 7.60 | 10.27 | 11.61 | 10.08 | 10.77 |

from the AlpacaCleaned dataset (Taori et al., 2023), with evaluations performed by GPT-4o-2024-08-06 (OpenAI et al., 2024). As shown in Table 9, we find that the task performance (length-controlled win rate, LCWR) continues to improve for a period even after the model's test perplexity surpasses its minimum and potentially increases. This sustained or improving task performance, despite non-optimal and potentially rising perplexity, aligns with the core characteristics of over-memorization, suggesting its relevance extends to generation-centric evaluations.

### E.3 ADDITIONAL EXPERIMENTS FOR DIFFERENT FINETUNING METHODS

This section provides more detailed experimental results than could be included in the main paper due to space constraints, expanding on the findings for both finetuning methods and model size scaling. For the various finetuning methods (PiSSA (Meng et al., 2024), LoRA+ (Hu et al., 2022), MiLoRA (Wang et al., 2025), and Full finetuning) discussed in Section 4.3, the main paper presented selected test set results (perplexity and accuracy) under various learning rates. This appendix offers a more comprehensive evaluation by presenting perplexity and accuracy data across a broader range of learning rates and by including results from the training and validation sets in addition to the test set. The results for the PiSSA method are shown in Figure 10, for the LoRA+ method in Figure 11, for the MiLoRA method in Figure 13, and for the Full finetuning method in Figure 14. Detailed descriptions of these methods and the observed over-memorization phenomena remain in Section 4.3.

Regarding the model size scaling experiments with Qwen2.5 models, introduced in Section E.1, the main paper included test set perplexity and accuracy for four learning rates, with figures typically visualizing results for one learning rate across different model sizes. To facilitate a clearer analysis of how each individual model's performance varies across these different learning rates, this appendix presents the test set results in an alternative format in Figure 15. In these figures, each subfigure is dedicated to a single model size (e.g., Qwen2.5-0.5B) and displays its performance across the aforementioned learning rates.

### E.4 ADDITIONAL EXPERIMENTS FOR CODE GENERATION

To complement the results presented in Section 4.4, which analyzed over-memorization on code generation tasks using LoRA (2e-4), we additionally finetune the LLaMA-3.1-8B model using more learning rate on the CodeFeedback 10K dataset and evaluate performance on HumanEval, as shown in Figure 12.

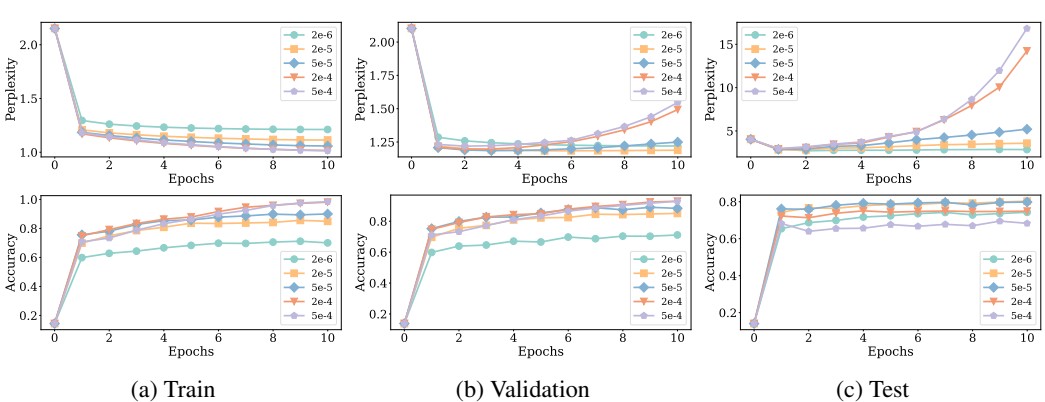

Figure 10: Evaluating LLMs finetuned by PiSSA with different learning rates, assessing train, validation, and test perplexity and accuracy.

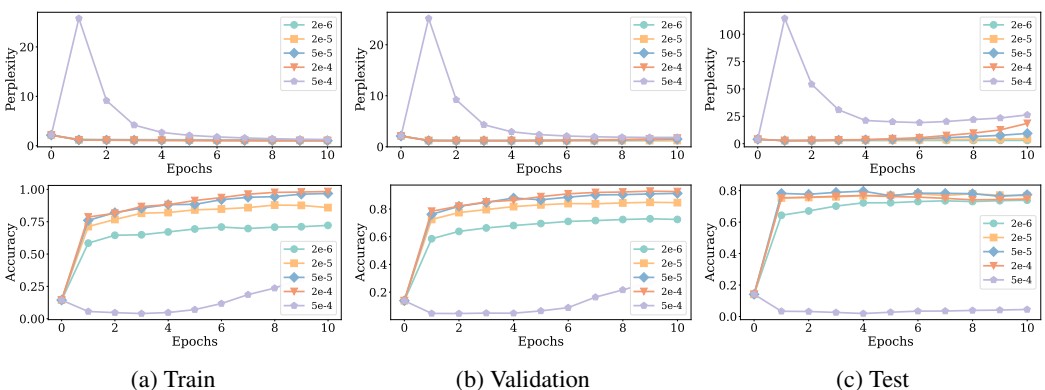

Figure 11: Evaluating LLMs finetuned by LoRA+ with different learning rates, assessing train, validation, and test perplexity and accuracy.

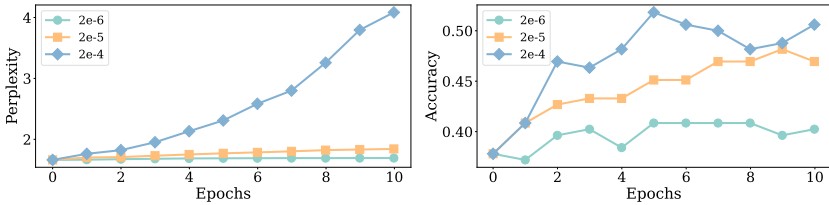

Figure 12: Test perplexity and accuracy curves illustrating over-memorization in HumanEval.

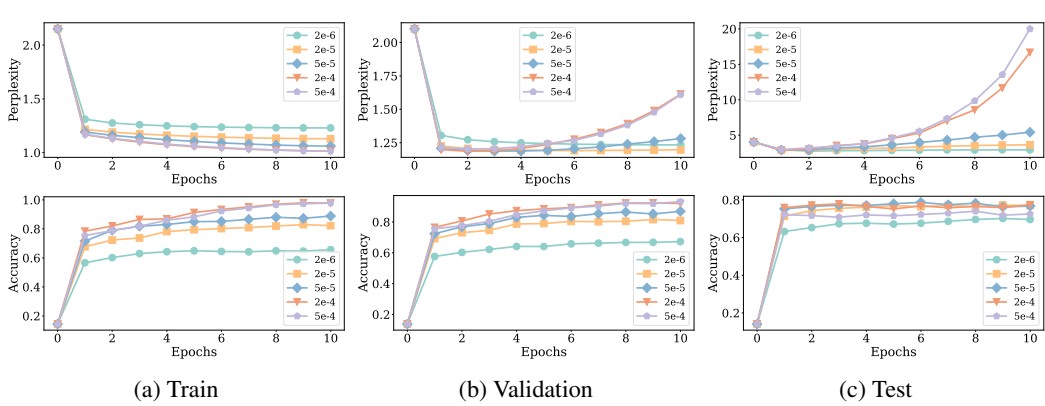

Figure 13: Evaluating LLMs finetuned by MiLoRA with different learning rates, assessing train, validation, and test perplexity and accuracy.

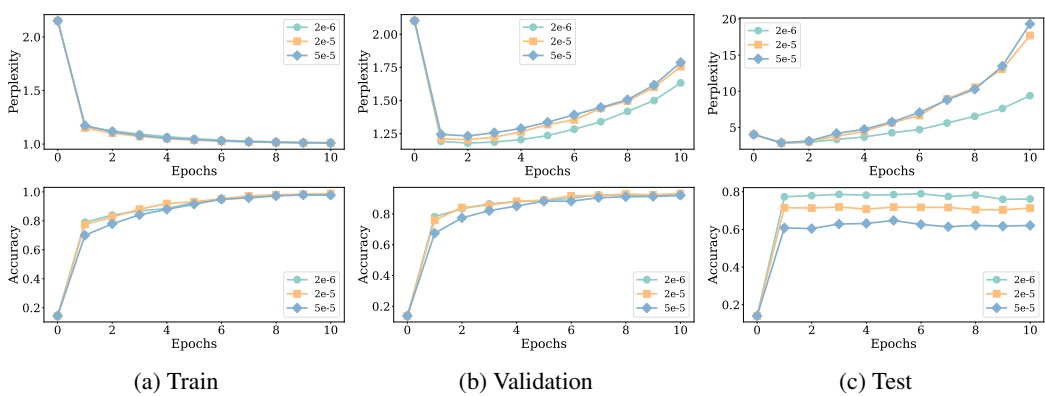

Figure 14: Evaluating LLMs fully finetuned with different learning rates, assessing train, validation, and test perplexity and accuracy.

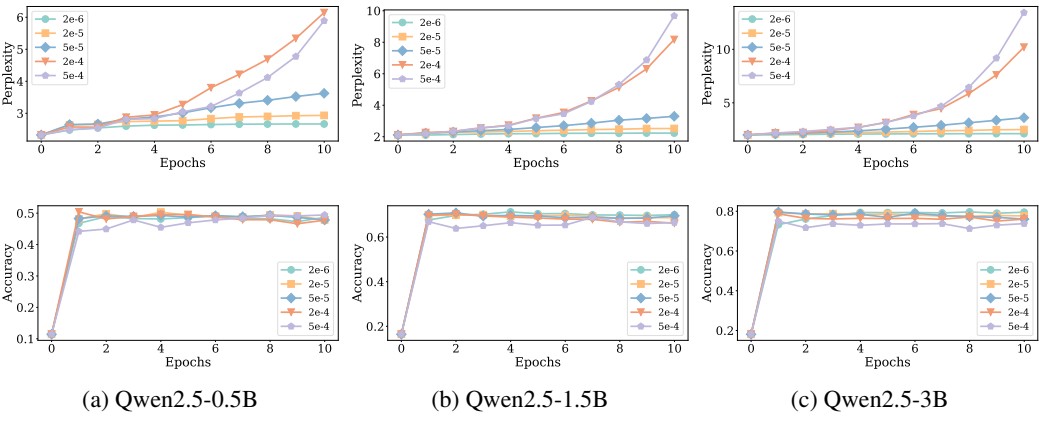

Figure 15: Evaluation of perplexity and accuracy on the test sets at different sizes of Qwen2.5 model.

