# OpenReview forum: "Unveiling Over-Memorization in Finetuning LLMs for Reasoning Tasks"
_ICLR.cc/2026/Conference — ICLR 2026 Conference Withdrawn Submission_

### Official Review · Reviewer_XDrx · 2025-10-24

**Soundness:** 2
**Presentation:** 2
**Contribution:** 2
**Rating:** 2
**Confidence:** 4

**Summary:**

This paper unveils the over-memorization phenomenon in the case of reasoning of LLM. The authors unveil the conclusion via conducting experiments on math datasets with three backbones and various lora variants. They try to propose a method to mitigate this problem.

**Strengths:**

1. The paper is well-written with clear storyline.
2. The paper's conclusion is clear, with experiments supported.

**Weaknesses:**

1. **Limited novelty and generalizability of the main conclusion**
   The central claim—that supervised fine-tuning (SFT) tends to memorize [2]—has been extensively discussed in prior work. The paper’s contribution would be stronger if it offered a more nuanced or novel perspective on this phenomenon.

   More critically, the experimental setup undermines the generalizability of the findings. The authors repeatedly fine-tune on a small dataset (e.g., 100K examples) for many epochs (e.g., 10), which is not representative of modern SFT practices. In real-world scenarios, large-scale reasoning datasets such as OpenR1 (220K), MetaMath (395K), or NuminaMath CoT (860K) are commonly used, often with only 1–3 epochs. Under such realistic conditions, the observed over-memorization may not occur, and the conclusions—and by extension, the proposed MAR method—may not hold. Without experiments on larger, more diverse datasets or with standard training protocols, the practical utility and robustness of MAR remain unconvincing.

2. **Concerns about the formulation and theoretical grounding of MAR**
   The proposed MAR loss is defined as
   \begin{equation}
   \mathcal{L} = \sum_{t=1}^T \big(1 - p_\theta(y_t \mid x, y_{<t})\big) \cdot \ell(y_t),
\end{equation}
   which is the inverse of the weighting scheme used in DFT [3] (where the weight is $p_\theta(y_t \mid x, y_{<t})$). While the intuition—down-weighting high-confidence tokens to mitigate memorization—is plausible, the paper lacks theoretical justification or empirical ablation to validate this design choice. In particular, it is unclear whether this inversion genuinely addresses over-memorization or merely shifts the optimization dynamics in a way that coincidentally improves test performance on narrow benchmarks. Robustness across architectures, datasets, and training regimes has not been demonstrated.

3. **Triviality of the analysis on over-memorization conditions**
   The paper claims to “explore the conditions that contribute to over-memorization” (Line 17), but the analysis appears superficial. For instance, the PPL on GSM8K test set (human-written reasoning traces) and MetaMath QA (GPT-3.5-generated traces) is attributed to over-memorization. However, this gap is more naturally explained by the distributional mismatch between $p_{\text{human}}$ and $p_{\text{GPT}}$: SFT aligns the model with the training distribution ($p_{\text{GPT}}$), so high PPL score on human-authored test data is expected—not surprising. Similarly, overfitting under excessive epochs, large learning rates, or increased model capacity is a well-known phenomenon in traditional supervised learning [1]. Framing these as novel insights into LLM fine-tuning overstates the contribution.



## Reference

[1] Smith L N. A disciplined approach to neural network hyper-parameters: Part 1--learning rate, batch size, momentum, and weight decay[J]. arXiv preprint arXiv:1803.09820, 2018.

[2] Chu T, Zhai Y, Yang J, et al. SFT Memorizes, RL Generalizes: A Comparative Study of Foundation Model Post-training[C]//Forty-second International Conference on Machine Learning.

[3] Wu Y, Zhou Y, Ziheng Z, et al. On the generalization of sft: A reinforcement learning perspective with reward rectification[J]. arXiv preprint arXiv:2508.05629, 2025.

**Questions:**

1. In Figures 2 and 4, Llama-3.1-8B exhibits more severe over-memorization compared to Mistral and Gemma under identical training settings. Could the authors hypothesize why this architecture is particularly prone to memorization? Is this due to architectural differences (e.g., RoPE vs. ALiBi, attention mechanisms), training data composition, or initialization?

---

> ### Author Response · Authors · 2025-11-23
> **Rebuttal 1**
>
> **1. Limited novelty and generalizability of the main conclusion.
> The central claim—that supervised fine-tuning (SFT) tends to memorize [2]—has been extensively discussed in prior work. The paper’s contribution would be stronger if it offered a more nuanced or novel perspective on this phenomenon. More critically, the experimental setup undermines the generalizability of the findings. The authors repeatedly fine-tune on a small dataset (e.g., 100K examples) for many epochs (e.g., 10), which is not representative of modern SFT practices. In real-world scenarios, large-scale reasoning datasets such as OpenR1 (220K), MetaMath (395K), or NuminaMath CoT (860K) are commonly used, often with only 1–3 epochs. Under such realistic conditions, the observed over-memorization may not occur, and the conclusions—and by extension, the proposed MAR method—may not hold. Without experiments on larger, more diverse datasets or with standard training protocols, the practical utility and robustness of MAR remain unconvincing.**
>
> **Response 1**
>
> We clarify the novelty and address the concern regarding the experimental setup.
>
>
> 1. Existing work on memorization in LLMs primarily focuses on training data memorization, often motivated by privacy or copyright concerns, or examines memorization dynamics during pretraining. These studies typically define memorization as a model's tendency to reproduce training samples and analyze how such behaviors affect generalization or data leakage.
>
> 2. Our study differs from this line of work by focusing on a distinct phenomenon we term over-memorization, which arises during the supervised fine-tuning phase. Unlike traditional overfitting, where continued training leads to increasing training accuracy but a clear drop in test accuracy, we observe that LLMs can reach a state where test perplexity increases while test accuracy remains high. This suggests that the model relies excessively on memorized solution patterns, which compromises its robustness, reduces output diversity, and weakens generalization to distribution shifts. To the best of our knowledge, this is the first comprehensive investigation of over-memorization as a distinct fine-tuning failure mode in LLMs.
>
> 3. Fine-tuning on partial datasets is common in practice. Recent work such as [1] shows that high-quality SFT often requires only small- to medium-scale datasets, and our 100K subset is already larger than many widely used SFT settings. The phenomenon also persists across dataset sizes, models, and fine-tuning algorithms, and it consistently appears once training continues for too long, regardless of scale. Even though some pipelines use 1–3 epochs on larger datasets, many real applications—especially academic or resource-limited ones—fine-tune on subsets or rely on validation accuracy alone, where this failure mode is more likely to occur. Our observations therefore reflect realistic usage rather than a corner case.
>
> 4. The main contribution of this paper is identifying and characterizing over-memorization. The mitigation techniques (MAR, Linear Merge) are intentionally lightweight and serve to show that once the failure mode is recognized, it can be alleviated without architectural changes or heavy computation. They are not intended to be the primary source of novelty; instead, they support the practical value of understanding this phenomenon.
>
> Overall, over-memorization is distinct from prior memorization concepts, is not restricted to small-scale settings, and has concrete implications for robustness and generalization in modern reasoning models.

---

> > ### Author Response · Authors · 2025-11-23
> > **Rebuttal 2**
> >
> > **2. Concerns about the formulation and theoretical grounding of MAR
> > The proposed MAR loss is defined as which is the inverse of the weighting scheme used in DFT [3] (where the weight is ). While the intuition—down-weighting high-confidence tokens to mitigate memorization—is plausible, the paper lacks theoretical justification or empirical ablation to validate this design choice.**
> >
> > **Response 2:**
> >
> > - First, the formulation of MAR is derived from DFT. DFT provides a principled derivation for weighting tokens based on their predicted probabilities. However, DFT assigns *higher weights* to tokens with *higher confidence*, assuming that confident tokens deserve more emphasis.  In contrast, our perspective differs fundamentally. After the first epoch of fine-tuning, the model has already "memorized" many training tokens. Rather than interpreting high probability as confidence, we treat it as an indicator of **memory**. Tokens with high predicted probability are therefore considered *already memorized*, and further emphasizing them contributes to over-memorization.
> > Under this memory-centric interpretation, we invert the DFT weighting formulation:   tokens with **higher memorization** receive **lower weight**, and tokens with **lower memorization** receive **higher weight**.
> > This directly follows from our analysis of the over-memorization phenomenon and serves as a simple mitigation strategy tailored to this issue.
> >
> > - Second, we emphasize that the main contribution of our paper lies in **identifying and analyzing over-memorization as a previously unrecognized training-stage regime** in LLM finetuning. MAR is intentionally presented as a lightweight auxiliary technique rather than the core novelty of the work.
> >
> > **3. Triviality of the analysis on over-memorization conditions
> > The paper claims to “explore the conditions that contribute to over-memorization” (Line 17), but the analysis appears superficial. For instance, the PPL on GSM8K test set and MetaMath QA is attributed to over-memorization. However, this gap is more naturally explained by the distributional mismatch between  human and gpt: SFT aligns the model with the training distribution, so high PPL score on human-authored test data is expected—not surprising. Similarly, overfitting under excessive epochs, large learning rates, or increased model capacity is a well-known phenomenon in traditional supervised learning [1]. Framing these as novel insights into LLM fine-tuning overstates the contribution.**
> >
> > **Response 3:**
> >
> > Over-memorization and traditional overfitting are related but distinct phenomena.
> >
> > - Traditional **overfitting** refers to a situation where a model continues to improve on the training data but begins to perform worse on the test set, typically resulting in **declining test accuracy**. This behavior is well-studied in traditional machine learning.
> >
> > - In contrast, we define **over-memorization** as a regime that commonly arises during the fine-tuning of **overparameterized and pretrained LLMs** in reasoning tasks, where the model continues to achieve **high test accuracy**, but its **test perplexity increases** significantly. This indicates that while the model still outputs the correct final answer, it becomes more rigid in its internal reasoning process, relying heavily on memorized patterns rather than diverse or adaptable reasoning paths.
> >
> > Our empirical analysis shows that over-memorized models:
> > - Exhibit **reduced output diversity**, making them less effective in test-time scaling (e.g., best-of-N decoding),
> > - Show **weaker robustness** to prompt variations and out-of-distribution generalization.
> >
> > **4. In Figures 2 and 4, Llama-3.1-8B exhibits more severe over-memorization compared to Mistral and Gemma under identical training settings. Could the authors hypothesize why this architecture is particularly prone to memorization? Is this due to architectural differences (e.g., RoPE vs. ALiBi, attention mechanisms), training data composition, or initialization?**
> >
> > **Response 4:**
> >
> > LLaMA-3.1-8B, Mistral-7B, and Gemma-2-9B differ at the same time in model architecture, pretraining corpora, tokenizer design, optimization schedule, and initialization. Because these factors vary together and **their pretraining datasets are not publicly available, it is extremely difficult to isolate which component contributes to the difference in over-memorization severity.**
> >
> > Given the number of uncontrolled variables, differences across model families are expected. Our intention is not to determine which model memorizes more strongly, but to demonstrate that all architectures, despite these differences, consistently exhibit the over-memorization pattern. This supports the claim that the phenomenon is general rather than specific to any particular backbone.

---

> > > ### Comment · Reviewer_XDrx · 2025-11-24
> > >
> > > I appreciate the authors' comments. However, some questions/concerns still remain:
> > > 1. What is [1] in your comment? I cannot find a reference here.
> > > 2. The authors claim that their setting is built upon fine-tuning on high-quality datasets with limited data. However, this claim unveils that their findings are restricted to this scenario, instead of broader, general supervised fine-tuning. This could be somewhat of an overclaim.
> > > 3. Unveiling this phenomenon is interesting, but its contribution may not be sufficient for ICLR. More valuable analysis (this corresponds to my question, as the authors do not give any explanation/additional analysis). For example, is this phenomenon common enough? You only select three dense model families with similar sizes (7-9B). Will this phenomenon exist in larger/smaller/MoE models? Your claim is that over-memorization is common for LLMs, so just these model backbones are not sufficient to validate this empirical finding.
> > > 4. The authors seem to omit my concern about the PPL drops on the GSM8K test set, which makes me worry that this is the point they fail to consider during their research.
> > >
> > > Given these concerns, I decided to remain my ratings.

---

> > > > ### Author Response · Authors · 2025-11-28
> > > > **Rebuttal 1**
> > > >
> > > > **1. What is [1] in your comment? I cannot find a reference here.**
> > > >
> > > > **Response 1：**
> > > >
> > > > - The reference “[1]” demonstrates that supervised fine-tuning can be highly effective even with **small- or medium-scale datasets**, which supports our choice of using a 100K-scale dataset.
> > > >
> > > > - In addition, recent work such as “[2]” shows that using only **11% of the MathInstruct training data** can already match the performance of the full dataset, and they further achieve strong results on the MATH benchmark with only **50,000 SFT examples**.
> > > >
> > > > - Moreover, the MiLoRA[3], PiSSA[4], and corda[5] adopt a nearly identical setting. MiLoRA and PiSSA are conducted on the **MetaMath-100K** dataset. This indicates that MetaMath-100K is a commonly used and reasonable benchmark size in recent empirical studies.
> > > >
> > > > We hope this clarifies the intended references and the rationale behind our dataset choice.
> > > >
> > > >
> > > > **2. The authors claim that their setting is built upon fine-tuning on high-quality datasets with limited data. However, this claim unveils that their findings are restricted to this scenario, instead of broader, general supervised fine-tuning. This could be somewhat of an overclaim.**
> > > >
> > > > **Response 2：**
> > > >
> > > > We clarify that our findings are not restricted to a narrow fine-tuning configuration, and we provide extensive evidence that the over-memorization phenomenon generalizes across data scales, domains, model architectures, and fine-tuning methods.
> > > >
> > > > **(1). Robustness across different data scales.** As shown in Appendix E.1, we compare datasets of varying sizes, and all of them consistently exhibit over-memorization. This indicates that the phenomenon is not dependent on a particular “small-data” setting, but appears across a meaningful range of data scales.
> > > >
> > > > **(2). Robustness across diverse reasoning domains.** Our analysis is not limited to mathematical reasoning. As shown in Section 4.4 and Appendix E.2, we evaluate the phenomenon on multiple heterogeneous domains, including:
> > > > - **HumanEval** (code reasoning)
> > > > - **GPQA** (scientific QA)
> > > > - **AlpacaEval 2.0** (open-ended instruction following; Appendix E.2)
> > > >
> > > > Over-memorization consistently occurs in all these domains, suggesting it is not tied to a specific task type.
> > > >
> > > > **(3). Robustness across different model architectures.** We evaluate multiple LLM families (e.g., Llama, Gemma, Mistral, Qwen), and all of them exhibit similar over-memorization trends. This demonstrates that the phenomenon is not an artifact of a particular architecture or pretraining recipe.
> > > >
> > > > **(4). Robustness across different fine-tuning methods.** Our experiments include several common fine-tuning strategies—such as LoRA, PiSSA, MiLoRA, and full fine-tuning—and over-memorization consistently appears across all of them. The convergence of results across methods highlights that the phenomenon is not tied to a specific fine-tuning algorithm.
> > > >
> > > > ---
> > > >
> > > > Overall, our experiments intentionally span **multiple data scales, diverse domains, different model architectures, and multiple fine-tuning methods**. While it is impossible to exhaustively enumerate all possible settings in practice, our broad and diverse evaluations strongly support the generality of the reported over-memorization phenomenon.

---

> > > > > ### Author Response · Authors · 2025-11-28
> > > > > **Rebuttal 2**
> > > > >
> > > > > **3. Unveiling this phenomenon is interesting, but its contribution may not be sufficient for ICLR. More valuable analysis (this corresponds to my question, as the authors do not give any explanation/additional analysis). For example, is this phenomenon common enough? You only select three dense model families with similar sizes (7-9B). Will this phenomenon exist in larger/smaller/MoE models? Your claim is that over-memorization is common for LLMs, so just these model backbones are not sufficient to validate this empirical finding.**
> > > > >
> > > > > **Response 3：**
> > > > >
> > > > > **(1). Value of empirical analysis:**  Empirical analysis plays an important role in advancing our understanding of how LLMs behave during fine-tuning. Our study contributes to this effort by identifying and characterizing *over-memorization*, a phenomenon that has not been documented previously yet is highly relevant to training and deployment.
> > > > >
> > > > > **(2). Generality across model families and scales:**  Our experiments extend well beyond a single backbone. As reported in Section 4.4 and Appendix E.1, we evaluate:
> > > > >
> > > > > - **Llama 3.1 8B**
> > > > > - **Mistral 7B v0.3**
> > > > > - **Gemma 2 9B**
> > > > > - **Qwen2.5 models (0.5B / 1B / 3B)**
> > > > >
> > > > > Across these **four model families and six model**, the same over-memorization behavior consistently appears. This suggests the phenomenon is not tied to a specific architecture or scale.
> > > > >
> > > > > **(3). Why the phenomenon arises:**  Supervised fine-tuning effectively minimizes a forward KL-equivalent objective:
> > > > >
> > > > > $$
> > > > > \min_{\theta} D\_{\text{KL}}(p, f\_\theta) \Longleftrightarrow \max_{\theta} \mathbb{E}\_{x \sim p(\cdot)} \mathbb{E}\_{y \sim p(\cdot | x)} [\log f\_\theta(y|x)].
> > > > > $$
> > > > >
> > > > > Forward KL is inherently mode-seeking: it places high probability on the single reference trajectory in the training data. In reasoning tasks with multiple valid solution paths, this encourages the model to over-specialize to the memorized path, reducing generative diversity and increasing perplexity.
> > > > >
> > > > >
> > > > > We evaluate six models across four families and multiple scales, all exhibiting the same pattern. Combined with the mode-seeking nature of the SFT objective, these results provide strong evidence that over-memorization is a common and general phenomenon in LLM fine-tuning.
> > > > >
> > > > > If the reviewer has further insights or theoretical directions, we would greatly appreciate the guidance and are eager to explore them in future work.
> > > > >
> > > > > **4.The authors seem to omit my concern about the PPL drops on the GSM8K test set, which makes me worry that this is the point they fail to consider during their research.**
> > > > >
> > > > > **Response 4：**
> > > > >
> > > > > **(1). Clarification of the PPL trend on GSM8K test set.** The perplexity on the GSM8K *test* set does **not** decrease during training. Instead, as shown in our results, the test-set PPL is:
> > > > >
> > > > > - **significantly higher** than the training and validation sets, and
> > > > > - exhibits a **decrease-then-increase** pattern as training progresses.
> > > > >
> > > > > This is fully consistent with our characterization of *over-memorization*: the model becomes increasingly confident in the memorized trajectories seen during training, which leads to higher perplexity when encountering diverse solution paths in the test set.
> > > > >
> > > > > **(2). Prior work also uses PPL to analyze memorization.** Using perplexity to measure how strongly a model memorizes specific reasoning trajectories is supported by prior work. For example, [6] employs PPL to quantify the model’s memorization of training-set solution paths and analyzes how training dynamics affect generalization.
> > > > >
> > > > > [1] Analyzing the Effects of Supervised Fine-Tuning on Model Knowledge from Token and Parameter Levels, EMNLP 2025.
> > > > >
> > > > > [2] SmallToLarge (S2L): Scalable Data Selection for Fine-tuning Large Language Models by Summarizing Training Trajectories of Small Models. NIPS 2024.
> > > > >
> > > > > [3] MiLoRA: Harnessing Minor Singular Components for Parameter-Efficient LLM Finetuning, NAACL 2025.
> > > > >
> > > > > [4] PiSSA: Principal Singular Values and Singular Vectors Adaptation of Large Language Models, NIPS 2024.
> > > > >
> > > > > [5] CorDA: Context-Oriented Decomposition Adaptation of Large Language Models for Task-Aware Parameter-Efficient Fine-tuning, NIPS 2024.
> > > > >
> > > > > [6] What Do Learning Dynamics Reveal About Generalization in LLM Mathematical Reasoning? ICML 2025.
> > > > >
> > > > > We hope this clarifies the test-set PPL behavior and its connection to over-memorization.

---

> > > > > > ### Comment · Reviewer_XDrx · 2025-11-28
> > > > > >
> > > > > > I appreciate the authors' further responses, which again resolved my concerns.
> > > > > > Due to the unintended bugs of OpenReview, I cannot increase my score further.
> > > > > >
> > > > > > But I would like to clarify that both MiLoRA and PiSSA conduct their experiments on the full MetaMathQA-395K dataset, instead of the 100K subset.
> > > > > >
> > > > > > If possible, I am willing to increase my score to 6. As for acceptance, I could only leave it to the Area Chair to determine.

---

### Official Review · Reviewer_7ccS · 2025-10-31

**Soundness:** 3
**Presentation:** 3
**Contribution:** 3
**Rating:** 4
**Confidence:** 4

**Summary:**

This paper investigates a practically important training-time regime in finetuning LLMs for reasoning tasks: test accuracy plateaus at a high level while test perplexity continues to increase. The authors document this regime across several model families (LLaMA, Mistral, Gemma), multiple adaptation methods (full FT, LoRA variants), and several task types, suggesting it is not an isolated artifact. They further probe downstream behaviors and show that models in this regime can exhibit consistent, though modest, degradation in OOD generalization, robustness, diversity, and privacy metrics, which implies that accuracy alone is an insufficient model-selection criterion. Finally, they discuss lightweight mitigations such as checkpoint selection/merging and a memorization-aware reweighting loss, although the latter needs clearer positioning relative to existing reweighting techniques.

**Strengths:**

1、Practical training takeaway. The paper highlights a realistic failure mode in finetuning pipelines: validation perplexity may start to rise while task accuracy is still improving, so stopping purely on perplexity can prematurely discard useful checkpoints. This makes the work directly useful to practitioners who fine-tune reasoning-capable LMs.
2、Multi-faceted behavioral probing. Beyond reporting the metric divergence, the authors systematically examine its downstream effects on OOD generalization, prompt robustness, diversity, calibration, and privacy, offering a broader picture of what this “high-accuracy, high-perplexity” phase might entail.
3、Strong reproducibility. The paper provides detailed dataset construction, model/configuration descriptions, and finetuning hyperparameters for several adaptation methods, making it straightforward for others to re-run, stress-test, or challenge the reported phenomenon.

**Weaknesses:**

1、Marginal Novelty of the Core Phenomenon: The paper's central concept of "over-memorization"—defined as rising test perplexity while test accuracy remains stable —is insufficiently distinguished from classical overfitting. The authors define classical overfitting as rising perplexity and decreasing accuracy, but the behavior they identify is arguably just a minor variant of this, where the task-specific metric (accuracy) is less sensitive or lags behind the loss metric (perplexity). The claim to be the "first to uncover the phenomenon"  is overstated.
2、Confounding Experimental Comparison: The behavioral analyses in Section 5 are based on a flawed comparison. The "normal model" is defined as the checkpoint from epoch 3, while the "over-memorized model" is the checkpoint from epoch 10 of the exact same training run. This is not a comparison of two different states; it is a comparison of an undertrained model against a fully trained model. A valid study would compare two models trained to convergence with different hyperparameters (e.g., one with a low learning rate, one with a high) that achieve the same high in-domain accuracy but different perplexity levels.
3、Significant Omission of Related Work for "MAR": The proposed "Memorization-Aware Reweighting" (MAR) mitigation technique appears to be a direct reimplementation of existing loss-weighting concepts, such as Focal Loss, which down-weight the loss contribution of easy (high-confidence) examples. The paper's formula $\mathcal{L}_{MAR}=\sum_{t=1}^{t}(1-p_{\theta}(y_{t}|x,y_{<t}))\cdot l(y_{t})$ is a clear example of this. The authors fail to cite, discuss, or compare their method to this extensive and highly relevant body of literature, which calls the novelty of this contribution into serious question.

**Questions:**

1、The paper sometimes describes the phenomenon as distinct from classical overfitting (no accuracy drop), but later aligns it with overfitting-like generalization issues. Could you make the taxonomy explicit: is over-memorization an early phase of overfitting in overparameterized LMs, or a genuinely different failure mode?
2、The current comparison (epoch 3 vs epoch 10 from the same run) may conflate undertraining with the proposed phenomenon. Please consider a matched-accuracy, different-training-condition comparison (e.g., low-LR-long vs high-LR-short) to isolate the effect.
3、Since the explanation relies on “multiple valid reasoning paths,” could you analyze generated chains from the two checkpoints to show they indeed follow different but correct trajectories?
4、MAR looks very close in spirit to focal-style reweighting. It would help to cite and position against this line of work, and ideally add a small comparison to show MAR is preferable in this setting.
Flag For Ethics Review

---

> ### Author Response · Authors · 2025-11-23
> **Rebuttal 1**
>
> **1. Marginal Novelty of the Core Phenomenon: The paper's central concept of "over-memorization"—defined as rising test perplexity while test accuracy remains stable —is insufficiently distinguished from classical overfitting. The authors define classical overfitting as rising perplexity and decreasing accuracy, but the behavior they identify is arguably just a minor variant of this, where the task-specific metric (accuracy) is less sensitive or lags behind the loss metric (perplexity). The claim to be the "first to uncover the phenomenon" is overstated.**
>
> **Response 1:**
>
> Over-memorization and traditional overfitting are related but distinct phenomena.
>
> - Traditional **overfitting** refers to a situation where a model continues to improve on the training data but begins to perform worse on the test set, typically resulting in **declining test accuracy**. This behavior is well-studied in traditional machine learning.
>
> - In contrast, we define **over-memorization** as a regime that commonly arises during the fine-tuning of **overparameterized and pretrained LLMs** in reasoning tasks, where the model continues to achieve **high test accuracy**, but its **test perplexity increases** significantly. This indicates that while the model still outputs the correct final answer, it becomes more rigid in its internal reasoning process, relying heavily on memorized patterns rather than diverse or adaptable reasoning paths.
>
> Our empirical analysis shows that over-memorized models:
> - Exhibit **reduced output diversity**, making them less effective in test-time scaling (e.g., best-of-N decoding),
> - Show **weaker robustness** to prompt variations and out-of-distribution generalization.
>
>
> **2. Confounding Experimental Comparison: The behavioral analyses in Section 5 are based on a flawed comparison. The "normal model" is defined as the checkpoint from epoch 3, while the "over-memorized model" is the checkpoint from epoch 10 of the exact same training run. This is not a comparison of two different states; it is a comparison of an undertrained model against a fully trained model. A valid study would compare two models trained to convergence with different hyperparameters (e.g., one with a low learning rate, one with a high) that achieve the same high in-domain accuracy but different perplexity levels.**
>
> **Response 2:**
>
> We appreciate the reviewer’s comment and would like to clarify why the comparison is valid.
>
> - **Epoch 3 as the "normal model"**. In most LLM fine-tuning studies, it is common to train for 3 epochs, as they have achieved reasonable convergence in accuracy. Epoch 3 in our setup represents such a model, which is neither undertrained nor fully overfitted.
>
> - **Comparison within the same training run**. Comparing epoch 3 and epoch 10 from the same training trajectory allows us to isolate the impact of **training duration** on the behavior of the model. The comparison focuses on **the same model**, under the same parameters, just at different stages of training. This helps us track how the model transitions from normal behavior to over-memorization.
>
> - **Training dynamics across hyperparameters**. In **Figure 2**, we show how LoRA performs with different learning rates (2e-5, 5e-5, and 2e-4) over training time. The same trends are observed across various learning rates in **Figures 10, 11, 13, and 14** for PiSSA, LoRA+, MiLoRA, and Full FT. These figures demonstrate that **higher learning rates** lead to **earlier onset of over-memorization**, while **lower learning rates** delay it. For example, with **2e-6**, LoRA struggles to achieve good performance, while **5e-4** causes unstable training. These cases highlight that **the over-memorization phenomenon is not simply due to prolonged training** but rather stems from the **learning rate settings**, as seen in our training dynamics figures.

---

> > ### Author Response · Authors · 2025-11-23
> > **Rebuttal 2**
> >
> > **3. Significant Omission of Related Work for "MAR": The proposed "Memorization-Aware Reweighting" (MAR) mitigation technique appears to be a direct reimplementation of existing loss-weighting concepts, such as Focal Loss, which down-weight the loss contribution of easy (high-confidence) examples. The paper's formula MAR is a clear example of this. The authors fail to cite, discuss, or compare their method to this extensive and highly relevant body of literature, which calls the novelty of this contribution into serious question.**
> >
> > **Response 3:**
> >
> > We clarify the relationship between our proposed Memorization-Aware Reweighting (MAR) and existing loss-weighting methods.
> >
> > - First, the formulation of MAR is derived from DFT. DFT provides a principled derivation for weighting tokens based on their predicted probabilities. However, DFT assigns *higher weights* to tokens with *higher confidence*, assuming that confident tokens deserve more emphasis.  In contrast, our perspective differs fundamentally. After the first epoch of fine-tuning, the model has already "memorized" many training tokens. Rather than interpreting high probability as confidence, we treat it as an indicator of **memory**. Tokens with high predicted probability are therefore considered *already memorized*, and further emphasizing them contributes to over-memorization.
> >
> > Under this memory-centric interpretation, we invert the DFT weighting formulation:
> > tokens with **higher memorization** receive **lower weight**, and tokens with **lower memorization** receive **higher weight**.
> > This directly follows from our analysis of the over-memorization phenomenon and serves as a simple mitigation strategy tailored to this issue.
> >
> > - Second, we emphasize that the main contribution of our paper lies in **identifying and analyzing over-memorization as a previously unrecognized training-stage regime** in LLM finetuning. MAR is intentionally presented as a lightweight auxiliary technique rather than the core novelty of the work.
> >
> > We appreciate the reviewer’s suggestion and will revise the paper to cite and discuss Focal Loss and related literature to properly acknowledge existing work in the broader family of loss-reweighting methods.
> >
> >
> > **4. The paper sometimes describes the phenomenon as distinct from classical overfitting (no accuracy drop), but later aligns it with overfitting-like generalization issues. Could you make the taxonomy explicit: is over-memorization an early phase of overfitting in overparameterized LMs, or a genuinely different failure mode?**
> >
> > **Response 4:**
> >
> > Over-memorization is a distinct training regime in overparameterized LLMs rather than an early phase of classical overfitting. In classical overfitting, prolonged training leads to a clear drop in test accuracy. In our experiments, even when training was extended to 50 epochs, test accuracy did not decline. The model instead became more rigid and memory-driven while still producing correct final answers. This indicates a different failure mode in which accuracy remains stable but the underlying reasoning distribution collapses, resulting in reduced diversity, weaker robustness, and poorer OOD performance.
> >
> > **5. The current comparison (epoch 3 vs epoch 10 from the same run) may conflate undertraining with the proposed phenomenon. Please consider a matched-accuracy, different-training-condition comparison (e.g., low-LR-long vs high-LR-short) to isolate the effect.**
> >
> > **Response 5:**
> >
> > - In most recent LLM fine-tuning works, training for 1–3 epochs is standard practice, and models at epoch 3 are generally considered converged rather than undertrained. This is also reflected in our results. Figure 2 for LoRA and Figures 10, 11, 13, and 14 for PiSSA, LoRA+, MiLoRA, and Full FT show that test accuracy reaches a high and stable level around epoch 3 across different learning rates. This indicates that epoch 3 is already a strong model and not an undertrained baseline.
> >
> > - We also compare different finetuning methods under the same learning rate in Figure 3. Figure 3(a) presents LoRA, LoRA+, PiSSA, MiLoRA, and Full Finetuning under 2e-6. Figures 3(b) and 3(c) show the same comparison under 2e-5 and 2e-4. These results demonstrate that over-memorization consistently appears across different finetuning strategies once training continues beyond the early epochs, even when the models achieve similar accuracy levels.

---

> > > ### Author Response · Authors · 2025-11-23
> > > **Rebuttal 3**
> > >
> > > **6. Since the explanation relies on “multiple valid reasoning paths,” could you analyze generated chains from the two checkpoints to show they indeed follow different but correct trajectories?**
> > >
> > > **Response 6:**
> > >
> > > - Figure 1 (right) already provides a concrete example showing that the two checkpoints exhibit clearly different token-level behaviors, where the over-memorized model assigns noticeably higher perplexity to several intermediate reasoning tokens. This reflects a shift in the internal reasoning trajectory despite producing the same final answer.
> > >
> > > - Additional evidence comes from Appendix D.3 and Table 8, which show that the over-memorized model has much lower output diversity. This further supports the interpretation that the model collapses onto a narrower set of reasoning paths, while the earlier checkpoint produces more varied and flexible chains.
> > >
> > > **7. MAR looks very close in spirit to focal-style reweighting. It would help to cite and position against this line of work, and ideally add a small comparison to show MAR is preferable in this setting. Flag For Ethics Review**
> > >
> > > **Response 7:**
> > >
> > > We will add the relevant discussion and citations to focal-style reweighting methods in the revised version. MAR is introduced as a simple mitigation specific to the over-memorization regime, and we will clarify its relation to this prior line of work.

---

> ### Author Response · Authors · 2025-11-28
>
> Dear Reviewer 7ccS,
>
> Thank you for your detailed and insightful review. We have added additional analyses to address all the questions you raised. If you have any further suggestions or would like clarification on any part of our response, please feel free to let us know. We would be very happy to continue the discussion.
>
> We truly appreciate the time and effort you have devoted to reviewing our work.
>
> Best regards,
>
> The Authors

---

### Official Review · Reviewer_GHrg · 2025-11-01

**Soundness:** 2
**Presentation:** 3
**Contribution:** 2
**Rating:** 4
**Confidence:** 4

**Summary:**

The paper reports an empirical phenomenon in LLM finetuning for reasoning tasks that the authors call over-memorization: after early training gains, test perplexity rises while test accuracy stays high, and this coincides with reduced robustness, OOD generalization, and generation diversity. The effect is shown across learning rates, epochs, finetuning methods (LoRA, PiSSA, full FT), and models; the authors also propose two mitigations—checkpoint merging and a memorization-aware reweighting (MAR) objective—and offer checkpoint-selection guidance.

**Strengths:**

Clear empirical pattern: Multiple plots/tables show rising test PPL without collapsing accuracy, across tasks and models.

Breadth: Results cover math QA, code, and scientific QA; also Gemma/Mistral.

Practical takeaways: Concrete checkpoint-selection advice (balance val-ACC with val-PPL) and evidence that it matters.

Robustness/OOD/Diversity analyses: Over-memorized checkpoints are more brittle to neutral prompt preambles and underperform on OOD.

Lightweight mitigations: Checkpoint merging (explicit formula) and MAR are simple and effective.

**Weaknesses:**

Causality vs. correlation. While learning rate and training time correlate with the effect, other confounds (batch size, data curriculum/order, decoding temperature during evaluation, regularization, LoRA rank, prompt templates) are not systematically ruled out. The methodological breadth is good, but ablations feel incomplete

Novelty positioning could be sharper. The paper claims to be the first to uncover this specific phenomenon; related work (e.g., learning-dynamics perspectives) is noted, but the boundary between “over-memorization” and known overfitting/under-determination behaviors is not crisply formalized. A precise decision rule for when a checkpoint is “over-memorized” is missing.

Significance of OOD/robustness deltas. OOD drops (~2 points on average) and perturbation losses are suggestive but modest; there are no confidence intervals or statistical tests, and some test sets are small/sensitive. Please report variance across seeds and significance for Table 2/Table 3.

**Questions:**

Could checkpoint merging be extended beyond linear two-point averaging (e.g., weight-space ensembling across more steps), and what are the failure cases?

For Table 2 and Table 3, please include CI/SE over seeds and explain whether differences are statistically significant.

---

> ### Author Response · Authors · 2025-11-23
> **Rebuttal 1**
>
> **1. Causality vs. correlation. While learning rate and training time correlate with the effect, other confounds (batch size, data curriculum/order, decoding temperature during evaluation, regularization, LoRA rank, prompt templates) are not systematically ruled out. The methodological breadth is good, but ablations feel incomplete.**
>
> **Response 1:**
>
> In our experiments, batch size, regularization, LoRA rank, and prompt templates were all fixed using standard and widely adopted configurations, following the settings in MiLoRA [1]. Exploring all combinations of these factors would lead to exponential computational cost and is not feasible for large-scale LLM studies. Most prior work also controls these variables for fairness.
>
> Our study already covers a wide range of training approaches such as LoRA, MiLoRA, PiSSA, PiSSA+, and full finetuning, as well as multiple model architectures including Llama, Mistral, Gemma, and Qwen. The consistency of results across these settings suggests that the observed effect is robust and not driven by a particular confounding factor.
>
> [1] MiLoRA: Harnessing Minor Singular Components for Parameter-Efficient LLM Finetuning. NAACL 2025.
>
>
> **2. Novelty positioning could be sharper. The paper claims to be the first to uncover this specific phenomenon; related work (e.g., learning-dynamics perspectives) is noted, but the boundary between “over-memorization” and known overfitting/under-determination behaviors is not crisply formalized. A precise decision rule for when a checkpoint is “over-memorized” is missing.**
>
> **Response 2:**
>
> Existing work on memorization in LLMs primarily focuses on training data memorization, often motivated by privacy or copyright concerns, or examines memorization dynamics during pretraining. These studies typically define memorization as a model's tendency to reproduce training samples and analyze how such behaviors affect generalization or data leakage.
>
> Our study differs from this line of work by focusing on a distinct phenomenon we term over-memorization, which arises during the supervised fine-tuning phase. Unlike traditional overfitting, where continued training leads to increasing training accuracy but a clear drop in test accuracy, we observe that LLMs can reach a state where test perplexity increases while test accuracy remains high. This suggests that the model relies excessively on memorized solution patterns, which compromises its robustness, reduces output diversity, and weakens generalization to distribution shifts. To the best of our knowledge, this is the first comprehensive investigation of over-memorization as a distinct fine-tuning failure mode in LLMs.

---

> > ### Author Response · Authors · 2025-11-23
> > **Rebuttal 2**
> >
> > **3. Significance of OOD/robustness deltas. OOD drops (~2 points on average) and perturbation losses are suggestive but modest; there are no confidence intervals or statistical tests, and some test sets are small/sensitive. Please report variance across seeds and significance for Table 2/Table 3.**
> >
> > **Response 3:**
> >
> > To address this concern, we conducted additional experiments using two standard seeds, 0 and 1000, in addition to the original seed 42. The results for both ID and OOD benchmarks, as well as prompt perturbation robustness, are reported below.
> >
> > **Table 1. Accuracy of different seeds and epochs on ID and OOD mathematical reasoning benchmarks**
> >
> > | Methods | GSM8K | MATH | SVAMP | ASDiv | MAWPS | TabMWP | Minerva | mmlu_math | OOD avg |
> > |-|-|-|-|-|-|-|-|-|-|
> > | seed 0 epoch3| 77.3| 29.5 | 77.3  | 81.9  | 91.8  | 69.8   | 29.0 | 16.0| 61.0|
> > | seed 0 epoch10| 77.6  | 29.4 | 75.9  | 80.1  | 89.9  | 64.5| 31.4| 17.1| 59.8|
> > | seed 42 epoch3|75.6| 28.9 | 77.1  | 81.7  | 89.7  | 67.1| 27.4| 17.9| 60.2|
> > | seed 42 epoch10| 76.4  | 12.0 | 75.3  | 79.1| 89.6| 63.6| 29.2| 12.0| 58.1|
> > | seed 1000 epoch3| 76.1| 29.0| 78.5  | 81.1  | 92.3  | 66.6| 29.8| 20.3| 61.4|
> > | seed 1000 epoch10  | 76.9| 29.3 | 75.8| 79.7| 88.3| 58.1| 30.0| 25.1| 59.5    |
> >
> > **Table 2. Accuracy under prompt perturbation for different seeds and epochs on GSM8K**
> >
> > | Prompt Type | seed 0 epoch3 | seed 0 epoch10 | seed 42 epoch3 | seed 42 epoch10 | seed 1000 epoch3 | seed 1000 epoch10 |
> > |-|-|-|-|-|-|-|
> > | first| 78.3| 76.9| 76.7| 74.7| 76.9| 75.8|
> > | today| 74.7| 71.4| 72.0| 70.3| 72.4| 72.6|
> > | we| 75.5| 72.4| 73.9| 72.0| 74.5| 72.3 |
> > | good | 73.5| 71.6| 73.6| 72.5| 73.7| 70.3|
> > | avg| 75.5| 73.1| 74.1| 72.4 74.4| 72.7|
> >
> > **Table 3. Statistical analysis across seeds**
> >
> > | Setting  | OOD avg | OOD SE | Prompt perturbation avg | Prompt perturbation SE |
> > |-|-|-|-|-|
> > | epoch 3  | 60.85   | 0.3750 | 74.64                    | 0.4446                 |
> > | epoch 10 | 59.15   | 0.5165 | 72.73                    | 0.2000                 |
> >
> > Across all three seeds, the magnitude of OOD drops and prompt perturbation effects remains consistent. The standard errors are small, indicating that the trends reported in Tables 2 and 3 of the paper are stable and not driven by randomness in initialization. We will include confidence intervals and standard errors in the revised version to provide a more complete statistical characterization.
> >
> >
> > **4. Could checkpoint merging be extended beyond linear two-point averaging (e.g., weight-space ensembling across more steps), and what are the failure cases?**
> >
> > **Response 4:**
> >
> > To investigate this, we conducted experiments on multiple merge strategies using the model trained with seed 0. The results are shown below:
> >
> > | Method                            | GSM8K | MATH | SVAMP | ASDiv | MAWPS | TabMWP | Minerva | mmlu_math | OOD avg |
> > |-|-|-|-|-|-|-|-|-|-|
> > | Epoch 3                           | 77.3  | 29.5 | 77.3  | 81.9  | 91.8  | 69.8   | 29.0    | 16.0      | 61.0    |
> > | Epoch 10                          | 77.6  | 29.4 | 75.9  | 80.1  | 89.9  | 64.5   | 31.4    | 17.1      | 59.8    |
> > | AVG Epoch 1–10                     | 81.7  | 33.4 | 78.6  | 82.6  | 92.2  | 72.1   | 31.4    | 19.7      | 62.8    |
> > | AVG Epoch 3 & 10                   | 80.2  | 31.1 | 79.3  | 82.1  | 92.1  | 69.2   | 32.8    | 18.2      | 62.3    |
> > | TA Epoch 3(0.3) & Epoch 10(0.7)   | 80.6  | 31.1 | 78.9  | 82.3  | 91.3  | 68.9   | 32.0    | 19.0      | 62.1    |
> > | TA Epoch 3(0.7) & Epoch 10(0.3)   | 78.4  | 30.8 | 78.6  | 82.1  | 92.5  | 71.6   | 32.0    | 17.7      | 62.4    |
> > | TIES Epoch 1–10                    | 45.3  | 16.2 | 68.1  | 62.0  | 71.2  | 47.3   | 16.8    | 36.0      | 50.2    |
> >
> > The experiments show that **merging checkpoints from all epochs (Epoch 1–10) achieves the best performance**, outperforming simple two-point merges such as Epoch 3 and Epoch 10. Weighted averaging (TA) with different ratios can also provide competitive results. In contrast, TIES, which zeros out part of the parameters, shows limited effectiveness.
> >
> > These results suggest that checkpoint merging **can be extended beyond linear two-point averaging**, and incorporating more steps generally improves performance. Methods that aggressively sparsify or zero parameters (like TIES) are less effective and represent a failure case.
> >
> > **5. For Table 2 and Table 3, please include CI/SE over seeds and explain whether differences are statistically significant.**
> >
> > **Response 5:**
> >
> > Following the suggestion, we have now included the standard error (SE) across three different seeds in Table 3 (see Response 3 above). These results quantify the variation and confirm that the observed differences in OOD accuracy and prompt perturbation robustness are statistically meaningful rather than random fluctuations. We will include SE in the final revision of Tables 2 and 3.

---

> ### Author Response · Authors · 2025-11-28
>
> Dear Reviewer GHrg,
>
> Thank you for your detailed and insightful review. We have added additional analyses to address all the questions you raised. If you have any further suggestions or would like clarification on any part of our response, please feel free to let us know. We would be very happy to continue the discussion.
>
> We truly appreciate the time and effort you have devoted to reviewing our work.
>
> Best regards,
>
> The Authors

---

### Official Review · Reviewer_Y3sc · 2025-11-11

**Soundness:** 3
**Presentation:** 2
**Contribution:** 2
**Rating:** 4
**Confidence:** 4

**Summary:**

The paper introduces and investigates an “over-memorization” phenomenon that emerges when fine-tuning large language models on reasoning tasks: test accuracy stays high while test perplexity keeps rising. Using LLaMA‑3.1‑8B on the MetaMathQA dataset, the authors show that larger learning rates accelerate this effect, whereas smaller rates eventually produce the same outcome. They propose two mitigation strategies—checkpoint merging and memory-aware reweighting—and demonstrate that both improve performance on in-domain and out-of-domain evaluations.

**Strengths:**

1. The paper systematically uncovers the “high accuracy, rising perplexity” over-memorization phenomenon during fine-tuning, verifies its prevalence across multiple learning rates and methods, and clearly distinguishes it from traditional overfitting.

2. It conducts a comprehensive experimental investigation that quantifies over-memorization’s negative impact on robustness, out-of-distribution generalization, Best-of-N sampling, and privacy risk with rich metrics.

3. The paper proposes practical mitigation strategies—checkpoint merging and memory-aware reweighting—that improve both ID and OOD performance, keep computational cost low, and are backed by ablation analyses for deployment guidance.

4. It maintains methodological rigor and reproducibility by detailing hyperparameters, data preprocessing, and multiple random seeds, while supplying full implementation specifics in the supplementary materials.

**Weaknesses:**

1. Lack of deep theoretical understanding: The paper’s central weakness is that it remains overly empirical. Its theoretical analysis of why over-memorization occurs is insufficient, offering only a superficial explanation via cross-entropy mechanics (Sec. 5). It provides no mathematical framework to predict when over-memorization will arise or to quantify its severity beyond empirical observation (Eqs. 3–4).

2. Questionable evaluation setup: Given the broad scope of the topic, comprehensive experiments are crucial. However, the study relies primarily on MetaMath for training and focuses on math reasoning benchmarks such as GSM8K and MATH. MetaMathQA was introduced in 2023, when GSM8K scores were still below 70; today, GSM8K and MATH (grade-school and high-school math) are largely saturated and offer limited insight. More challenging math evaluations—e.g., AIME 2024/2025—should be included. In addition, the paper lacks verification on other reasoning domains (logical, commonsense, etc.; Sec. 4.1) and evaluates only a single backbone (LLaMA‑3.1‑8B), which limits generality; models like the Qwen series should be tested to support broader claims.

3. Inconsistent and unclear mathematical formulations: Equation (2)’s memory-aware reweighting loss \(L_{\text{MAR}}\) lacks sufficient theoretical grounding for the chosen weighting scheme, and Equation (1)’s checkpoint merging uses a fixed 1/2 coefficient without theoretical or empirical justification.

**Questions:**

See “Weaknesses.” and consider the following advices：

Expand experimental scope and introduce statistical validation: The evaluation should be extended to diverse tasks such as logical reasoning (e.g., LogiQA), commonsense reasoning (e.g., CommonsenseQA), and reading comprehension, and the claims should be validated across multiple model scales (1B, 3B, 7B, 13B+) and architectures (Mistral, Gemma, Qwen, etc.) to strengthen generality.


I genuinely find this topic an interesting angle. However, the experimental analysis is currently too empirical; if the concerns above are thoroughly addressed, I would consider raising my score.

---

> ### Author Response · Authors · 2025-11-23
> **Rebuttal 1**
>
> **1. Lack of deep theoretical understanding: The paper’s central weakness is that it remains overly empirical. Its theoretical analysis of why over-memorization occurs is insufficient, offering only a superficial explanation via cross-entropy mechanics (Sec. 5). It provides no mathematical framework to predict when over-memorization will arise or to quantify its severity beyond empirical observation (Eqs. 3–4).**
>
> **Response 1:**
>
> Empirical evaluations are crucial for advancing the understanding of LLMs. While primarily empirical, our study identifies and analyzes over-memorization in LLM fine-tuning. We also provide below a conceptual explanation to illuminate its underlying cause.
>
> - **Optimization landscape of LLM**.Differ from traditional small-scale models which train from scratch, large-scale LLMs are extensively pretrained on diverse datasets. This pretraining creates a distinct optimization landscape for LLMs, which may contribute to the phenomenon of over-memorization.
>
> - **Training objective of supervised fine-tuning**. Supervised Fine-Tuning (SFT) typically minimizes cross-entropy loss over prompt–response pairs, which corresponds to minimizing the forward KL divergence between the empirical data distribution and the model distribution:
>
> We propose the following objective function:
>
> $$
> \min_{\theta} D\_{\text{KL}}(p, f\_\theta) \Longleftrightarrow \max_{\theta} \mathbb{E}\_{x \sim p(\cdot)} \mathbb{E}\_{y \sim p(\cdot | x)} [\log f\_\theta(y|x)].
> $$
>
> This objective is inherently mode-seeking, encouraging the model to concentrate probability mass on the observed outputs while ignoring other plausible alternatives [1]. In tasks such as mathematical reasoning, a single input can have multiple valid solution paths, yet training data typically contains only one reference trajectory. The model becomes overconfident in the memorized reasoning steps and less likely to explore alternative valid completions. This causes the model to fit to that specific path, reducing generative diversity and increasing test-time perplexity, even if the final answer remains accurate.
>
> If the reviewer has further insights or theoretical directions, we would greatly appreciate the guidance and are eager to explore them in future work.
>
>
>
> **2. Questionable evaluation setup: Given the broad scope of the topic, comprehensive experiments are crucial. However, the study relies primarily on MetaMath for training and focuses on math reasoning benchmarks such as GSM8K and MATH. MetaMathQA was introduced in 2023, when GSM8K scores were still below 70; today, GSM8K and MATH (grade-school and high-school math) are largely saturated and offer limited insight. More challenging math evaluations—e.g., AIME 2024/2025—should be included. In addition, the paper lacks verification on other reasoning domains (logical, commonsense, etc.; Sec. 4.1) and evaluates only a single backbone (LLaMA‑3.1‑8B), which limits generality; models like the Qwen series should be tested to support broader claims.**
>
> **Response 2:**
>
>
> We clarify that our evaluation is broader than it may appear.
>
> **1. Diverse reasoning domains.**
> Section 4.4 and Appendix E already evaluates multiple non-math tasks, including:
> - **HumanEval** (code reasoning),
> - **GPQA** (scientific QA),
> - **AlpacaEval 2.0** (open-ended generation; Appendix E.2).
> Across all these domains, the same over-memorization pattern is observed, showing that the phenomenon is not limited to mathematical reasoning.
>
> **2. Multiple model families and Data Size**
> Beyond LLaMA-3.1-8B, Section 4.4 and Appendix E.1 includes:
> - **Mistral-7B-v0.3**, **Gemma-2-9B**, **Qwen2.5 models (0.5B/1B/3B)**.
>
> - Appendix E.1 compares datasets of different scales.
> All exhibit consistent over-memorization behavior, demonstrating architectural and scale generality.
>
> **3. Dataset and benchmark coverage.**
>
>
> - MetaMathQA remains one of the most widely used SFT datasets for reasoning, and is appropriate for studying finetuning dynamics.
> - Beyond GSM8K and MATH, Section 5.1 and Table 2 include **six additional OOD reasoning benchmarks**:
> SVAMP, ASDiv, MAWPS, TabMWP, Minerva, and MMLU-STEM.
> These cover diverse numerical, symbolic, and STEM reasoning settings.
>
> Our experiments span multiple domains (math, code, science, open-ended), multiple model families (LLaMA, Mistral, Gemma, Qwen), and a wide range of ID/OOD benchmarks. Thus, the evidence supporting the generality of over-memorization is broad and robust. We will clarify this in the revision.

---

> > ### Author Response · Authors · 2025-11-23
> > **Rebuttal 2**
> >
> > **3. Inconsistent and unclear mathematical formulations: Equation (2)’s memory-aware reweighting loss (MAR) lacks sufficient theoretical grounding for the chosen weighting scheme, and Equation (1)’s checkpoint merging uses a fixed 1/2 coefficient without theoretical or empirical justification.**
> >
> > **Response 3:**
> >
> > We thank the reviewer for the observation and clarify the reasoning behind both formulations.
> >
> > - The MAR objective is based on the confidence-reweighting loss used in DFT, which provides a formal derivation for token-level weighting during finetuning. DFT increases the weight of high-confidence tokens, while in our setting these high probabilities after the first epoch correspond to tokens the model has already memorized. Our work studies finetuning through a memory perspective, so we invert the DFT weighting: tokens with high predicted probability are treated as already learned and receive lower weight, and low-probability tokens receive higher weight. This preserves the theoretical basis of DFT while adapting it to the dynamics of over-memorization.
> >
> > - The 1/2 coefficient in checkpoint merging follows the standard linear parameter averaging widely used in existing LLM merging practice. Public toolkits such as mergekit (https://github.com/arcee-ai/mergekit) list linear averaging as the default merging option, and many prior works rely on the same formulation. Our objective is not to introduce a new merging algorithm but to show that even the simplest commonly used averaging method is already effective in mitigating over-memorization. Empirically, this simple choice improves both ID accuracy and OOD robustness without requiring additional hyperparameters.

---

> ### Author Response · Authors · 2025-11-28
>
> Dear Reviewer Y3sc,
>
> Thank you for your detailed and insightful review. We have added additional analyses to address all the questions you raised. If you have any further suggestions or would like clarification on any part of our response, please feel free to let us know. We would be very happy to continue the discussion.
>
> We truly appreciate the time and effort you have devoted to reviewing our work.
>
> Best regards,
>
> The Authors

---

### Note · Authors · 2025-12-28

I have read and agree with the venue's withdrawal policy on behalf of myself and my co-authors.